# Modeling Total Electron Content derived from radio occultation measurements by COSMIC satellites over the African region

Patrick Mungufeni[1,2], Sripathi Samireddipalle[3], Yenca Migoya-Orué[4], and Yong Ha Kim[1]

[1]Department of Astronomy and Space Science, Chungnam National University, Daejeon, South Korea

[2]Physics department, Mbarara University of Science and Technology, P.O. Box 1410 Mbarara, Uganda

[3]Indian Institute of Geomagnetism, New Panvel, India

[4]The Abdus Salam International Centre for Theoretical Physics (ICTP) T/ICT4D

## Abstract

This study developed a model of Total Electron Content (TEC) over the African region. The TEC data were obtained from radio occultation measurements done by the Constellation Observing System for Meteorology, Ionosphere, and Climate (COSMIC) satellites. Data during geomagnetically quiet time (Kp < 3 and $Dst$ > -20 nT) for the years 2008 - 2011, and 2013 – 2017 were binned according to local time, seasons, solar flux level, geographic longitude and latitude. B splines were fitted to the binned data to obtain model coefficients. The model was validated using actual COSMIC TEC data of the years 2012 and 2018. The validation exercise revealed that, approximation of observed TEC data by our model produces root mean squared error of 5.02 TECU. Moreover, the modeled TEC data correlated highly with the observed TEC data (r = 0.93). Due to the extensive input data and the applied modeling technique, we were able to reproduce the well-known TEC features such as local time, seasonal, solar activity cycle, and spatial variations over the African region. Further validation of our model using TEC measured by ionosonde stations over South Africa at Hermanus, Grahamstown and Louisville revealed r values > 0.92 and RMSE < 5.56 TECU. These validation results imply that our model can estimate fairly well TEC that would be measured by ionosondes over locations which do not have the instrument. Another

importance of this study is the fact that it has shown the potential of using basis spline functions for modeling ionospheric parameters such as TEC over the entire African region.

## 1. Introduction

Among the error sources that affect the positioning in Global Navigation Satellite Systems (GNSS) are the propagation medium related errors. In particular, the ionospheric refraction is the largest contributor of the user equivalent range error. This type of frequency dependent error can virtually be eliminated in dual frequency receivers by differential techniques (Hofmann-Wellenhof et al., 2007). For the case of single frequency receivers, some GNSS (e.g Global Positioning System (GPS) and Galileo) broadcast message includes the parameters of an ionospheric model which can be used to compute and correct the ionospheric effects (Guochang, 2007). For instance, the GPS uses the Klobuchar model which represents the zenith delay as a constant value at night and a half cosine function during the day (Klobuchar, 1987). In the framework of the European Galileo constellation, the NeQuick G based on NeQuick model has been proposed to be used for single frequency positioning (see Issue 1.2, September, 2016 of European Commission, titled, European GNSS (Galileo) Open Service - Ionospheric correction algorithm for Galileo single frequency users). The NeQuick and its subsequent modifications (NeQuick G and NeQuick 2) are a three-dimensional, time dependent ionospheric electron density model developed by the Aeronomy and Radio Propagation Laboratory (ARPL) of the Abdus Salam International Center for Theoretical Physics (ICTP) in Trieste, Italy and the Institute for Geophysics, Astrophysics and Meteorology of the University of Graz, Austria (Nava et al., 2008). In addition to using models to reduce ionospheric refraction errors, Space Based Augumentation Systems (SBAS) such as the Wide Area Augmentation System (WAAS), the European Geostationary Navigation Overlay Service (EGNOS), and the GPS-aided Geo Augmented Navigation (GAGAN) are also used (Hofmann-Wellenhof et al., 2007).

For the international standard specification of ionospheric parameters (such as electron density, electron and ion temperatures, and equatorial vertical ion drift), the Committee

on Space Research (COSPAR) and the International Union of Radio Science (URSI) recommended    the International Reference Ionosphere Model (IRI) (Bilitza, 2001). IRI is an empirical model primarily based on all available experimental data (ground and space based) sources. However, theoretical considerations have been used in bridging data gaps and for internal consistency checks (Bilitza, 2001).

The ionospheric Total Electron Content (TEC) is one of the important descriptive physical quantities of the ionosphere (Rama Rao et al., 1997; Ercha et al., 2012). The GNSS measurements obtained from the global and regional networks of International GNSS Service (IGS) ground receivers have become a major source of TEC data. As one of the IGS analysis centers, Center for Orbit Determination in Europe (CODE) provides Global Ionosphere Maps (GIMs) containing vertical TEC data daily using the GNSS data collected from over 200 tracking stations of IGS and other institutions. Several studies have used GIMs from CODE and other IGS analysis centers such as the Jet Propulsion Laboratory (JPL) to construct TEC models (Jakowski et al. 2011a; Mukhtarov et al. 2013; Ercha et al. 2012; Sun et al., 2017). Jakowski et al. (2011a) proposed the Global Neustrelitz TEC Model (NTCM-GL) that describes the average TEC under quiet geomagnetic conditions. The NTCM-GL was developed using GIMs during 1998 - 2007 provided by CODE. A global background TEC model was also built using CODE GIMs by Mukhtarov et al. (2013). The model describes the climatological behavior of the ionosphere. The GIMs from JPL were used by Ercha et al. (2012) to construct a global ionosphere model using Empirical Orthogonal Function (EOF) analysis method. The Taiwan Ionosphere Group for Education and Research constructed a global ionosphere model from GNSS and the Constellation Observing System for Meteorology, Ionosphere, and Climate (COSMIC) GPS radio occultation (RO) observations (Sun et al., 2017). The map of all the averaged Root Mean Squared (RMS) error values of CODE GIMs during the years 2010 - 2012 presented by Najman and Kos (2014) showed high values over low latitude African regions. This could be due to the poor distribution of IGS tracking stations over Africa and inability of the spherical harmonics function used in GIM to describe ionospheric structure over low latitudes.

In addition to the existing GIMs discussed in the previous paragraph, regional TEC maps and models have also been constructed. In comparison with the global models, regional TEC models might have better accuracy over the particular region for which it was constructed. Opperman (2008) stated that the higher time and spatial resolution imaging achievable with regional models permits the analysis of localized ionospheric structures and dynamics not observable in global models. Examples of studies that developed TEC models over some parts of Africa are the following. A neural network model of GNSS - vertical TEC (GNSS-VTEC) over Nigeria was developed by Okoh et al., (2016) using all available GNSS data from the Nigerian GNSS Permanent Network (NIGNET). An adjusted spherical harmonic-based TEC model was developed by Opperman, (2008) using a network of South African dual frequency GPS receivers. Habarulema et al., (2011) presented the Southern Africa TEC prediction (SATECP) model that was based on the Neural Network technique. The SATECP generates TEC predictions as function of input parameters, namely, local time, day number of the year, solar and magnetic activity levels, and the geographical location. A neural network based ionospheric model was developed using GPS-TEC data over the East African sector by Tebabal et al. (2019). Recently, Okoh et al., (2019) used neural network technique to develop TEC model over the entire African region. In addition to using TEC obtained by COSMIC RO technique, they used TEC measured by GPS receivers on ground.

Due to the lack of a dense network of ground-based GNSS receivers and poor coverage of COSMIC RO data over the African region, the TEC model over the entire African region presented by Okoh et al. (2019) sometimes failed to capture the equatorial ionization anomaly (EIA) over the region. This point has been illustrated with examples in sections 2 and 5. In this study, we applied data binning method to the COSMIC RO TEC data that allowed development of an improved TEC model over the region. Moreover, we demonstrate the potential of the basis spline functions to model TEC over the African region. These basis functions never vanish over limited intervals and add up to one at all local times and longitudes (De-Boor, 1978). Moreover, according to Scherliess and Fejer, (1999), they are ideally suited to model the equatorial

ionosphere which exhibit smooth and rapid changes during daytime and near sunset, respectively, by proper placement of the mesh of nodes. In section 2, the data and methods of analysis that were used in the study are described. The details of the model proposed in this study are described in section 3. We present comparison between the observed and modeled TEC in section 4. The model validation and the conclusions are presented in sections 5 and 6, respectively.

## 2. The Data and methods

## 2.1 Data sources

In order to overcome the problem of lack of a dense network of ground based GNSS receivers over the African region, this study used TEC data obtained from RO measurements done by the COSMIC satellites. The integrated electron density (integration being done up to the altitudes of the COSMIC satellites) which is being referred to as TEC in this study can be obtained from ionPrf files which are processed at the COSMIC Data Analysis and Archive Centre (CDAAC)(http://cosmic-io.cosmic.ucar.edu/cdaac/index.html). The TEC for the individual occultation events were assigned to the geographic coordinates of NmF2 in the same file.

In order to get integrated electron density approximately up to the altitudes of GPS satellites, Okoh et al., (2019) used neural networks to learn the relationship between coincident TEC measurements done by ground based GPS receivers and COSMIC RO. They showed that the ratio between TEC data from the two sources vary spatially. This observation implies that the neural networks may not learn very well the relationship between TEC measured by ground-based GPS receivers and COSMIC RO over locations which do not have the former data set during the entire study period. As it can be seen in Figure 1 of Okoh et al., (2019), there were large spatial coverage's that do not have ground-based GPS receivers. Unlike what has been done in Okoh et al., 2019 and Mungufeni et al., 2019, in the current work we used only COSMIC TEC without any adjustments.

In this regard, an analysis of coincident ground-based GNSS TEC and TEC from COSMIC occultation data performed by Mungufeni et al. (2019) reveals that the upper quartile of the differences between the two data sets may reach up to ~11 TECU over the northern crest of the Equatorial Ionization Anomaly. Over the southern mid-latitude region, the differences were low (~4 TECU). Since the upper quartiles of the differences can reach up to ~11 TECU, the median/mean values in the worst cases might obviously be much lower than this value. This might be the reason for observing most of the well-known ionospheric TEC features over the African region when the COSMIC RO TEC were appropriately binned as in Mungufeni et al. (2019). Therefore, this study used the TEC obtained from COSMIC occultation measurements to develop TEC model over the African region in order to reproduce these ionospheric features. Such endeavors are important for educational purposes.

During geomagnetic storms, the variations in zonal electric fields and composition of the neutral atmosphere contribute significantly to the occurrence of negative and positive ionospheric storm effects in the low latitude region (Rishbeth and Garriot, 1969; Buonsanto, 1999; Adewale et al., 2011). Therefore, since the ionosphere changes in a complex manner during geomagnetic storms, we only considered data on quiet days. The quiet geomagnetic days were identified by examining the 3 hourly Kp and Disturbance storm time (Dst) indices that were obtained from the World Data Center of Kyoto, Japan (http://swdcwww.kugi.kyoto-u.ac.jp/). A day was considered to be quiet if all the 8 Kp values in that day were ≤ 3. In addition to satisfying this condition, the hourly values of Dst in that day should also have values ≥ -20 nT. The two conditions were applied to ensure that both low and mid/sub-auroral latitude geomagnetic disturbances are detected by Dst and Kp indices, respectively. In future, we intend to use TEC data during disturbed geomagnetic conditions to construct a TEC model during geomagnetically disturbed conditions.

## 2.2 Methods of Data Analysis

The TEC data during the years 2008 - 2011 and 2013 - 2017 were used for developing

the TEC model over the African region. Due to the adequate data needed to develop an empirical model, we only reserved the data of the years 2012 and 2018 for validation. The period considered in this study represents data of both low and high solar activity level in sunspot cycles 23 and 24. The data within geographic latitude and longitude ranges of -35 – 35$^o$ and -20 – 60$^o$, respectively, were used to cover the African region. Table 1 presents the number of days per year when there were TEC data over the African region. Since there are many geomagnetically disturbed days in high (2012 - 2015) and medium (2011 and 2016) solar activity years, the number of days with data is also reduced in such years compared to low solar activity years (2008 - 2010, 2018).

Table 1: Distribution of number of days with data

| Year | Number of days with data |
|------|--------------------------|
| 2008 | 219 |
| 2009 | 293 |
| 2010 | 235 |
| 2011 | 174 |
| 2012 | 169 |
| 2013 | 185 |
| 2014 | 164 |
| 2015 | 128 |
| 2016 | 151 |
| 2017 | 154 |
| 2018 | 211 |

It would be good to bin the TEC data according to geomagnetic latitudes since many structural and dynamical features of the ionized and neutral upper atmosphere are strongly organized by the geomagnetic field (e.g. Emmert et al., 2010). This may be complicated since geomagnetic latitude lines are not usually straight. For convenience and simplicity, we binned the data based on geographic coordinates. In order to observe small scale ionospheric structures, small grid resolutions of 3 and 5 degrees in

geographic latitude and longitude, respectively were used to bin the TEC data. These grid resolutions resulted into 24 and 16 latitudinal and longitudinal bins, respectively. Several studies (e.g. Krankowski et al., 2011 and Mengist et al., 2019) that have used COSMIC data commonly consider measurements with horizontal smear > 1500 km prone to errors and they reject such measurements. We established that after applying this restriction, there were ~40 RO measurements per day during the year 2013 over our study area (not shown here). Based on the previous discussions, this value is far less than the 9,216 (16 longitudinal, 24 latitudinal, and 24 local time) TEC data points required in all grid cells in a day. As stated in section 1, this poor amount of data to represent day of year TEC variation might be the reason for the failure of TEC model presented by Okoh et al. (2019) to capture in some cases the EIA over the African region. Another reason might be the discrepancy which arises due to some locations being represented by adjusted COSMIC RO TEC while others by the ground-based GPS TEC data.

Since empirical modeling requires adequate data for the mathematical functions to capture the physics inherent in the data, this study did not reject COSMIC RO TEC measurements with horizontal smear > 1500 km. Although not presented here, we observed that the COSMIC TEC data values with smear > 1500 km did not introduce alarming errors. This observation was made when we analyzed COSMIC TEC data which were coincident with TEC observed by ionosonde stations over South Africa (see details in section 5.2) located at Hermanus, Grahamstown, and Louisvale. Interestingly, compared to measurements with horizontal smear > 1500 km, some measurements with horizontal smear < 1500 km were observed to be far from the linear least squares fitting line. Further analysis of COSMIC RO observations over our study area revealed that without restricting horizontal smear, there were ~80 RO measurements per day during the year 2013 (not shown here). Still this value is far less than the 9,216 TEC data values required to fill all spatial grid cells in a day. To partially solve this problem, instead of binning data according to year, we binned the data according to different solar flux levels as shown below.

For each spatial grid cell, the data were binned at 1-hour interval. TEC values within the bins were averaged to yield 1-hour resolution TEC data over the grids. TEC data for the different days were binned according to F10.7 flux of that day. The F10.7 flux indices were obtained from the Space Weather Prediction Center (SWPC) of the National Oceanic and Space Administration (NOAA) (http://www.swpc.noaa.gov/). The F10.7 flux ranges for low solar activity (LSA), medium solar activity (MSA), and high solar activity (HSA) were < 76, 76 - 108, and > 108 sfu, respectively. The boundary values 76 and 108 sfu of the F10.7 flux ranges correspond to the $75th$ and $25th$ percentiles of all F10.7 flux values on the days in low (2008 - 2010, 2017 -2018) and high (2012 - 2015) solar activity years, respectively.

Table 2**:** Average monthly F10.7 flux values used in the study

| Month | F10.7 flux (sfu) | | |
|---|---|---|---|
| | LSA | MSA | HSA |
| January | 71.10 | 83.94 | 140.65 |
| February | 71.14 | 87.06 | 126.23 |
| March | 69.81 | 85.40 | 130.98 |
| April | 71.02 | 86.09 | 130.46 |
| May | 70.29 | 90.59 | 123.80 |
| June | 69.51 | 89.91 | 118.73 |
| July | 68.09 | 88.14 | 128.92 |
| August | 67.45 | 85.46 | 114.53 |
| September | 69.20 | 86.34 | 122.98 |
| October | 70.06 | 81.88 | 131.50 |
| November | 71.66 | 82.40 | 142.95 |
| December | 70.82 | 82.97 | 142.72 |

The data within a specific solar flux bin were further binned based on months of a year. The average of the corresponding F10.7 flux of the days used to represent seasonal TEC were determined and used to capture the variation of TEC with solar flux. Table 2 presents the average F10.7 flux values that were determined in the months of a year. In summary, a total of 331,776 TEC data values were needed to exist in 16 longitudinal, 24 latitudinal, 3 solar flux, 12 monthly, and 24 hourly bins, in order to determine the model coefficients. However, from the data of the entire study period, only 121,447 bins were filled with TEC data values. The average of the standard deviations of the bins that contained more than 1 TEC data during low (sample size = 21,108), medium (sample size = 6,180) and high (sample size = 7,495) solar flux levels were 1.28, 2.15, and 4.31 TECU, respectively.

The bins which did not have TEC data were filled by estimation following the procedures described in 3 steps below.

1. At a particular spatial grid cell, the diurnal TEC was divided into two local time sectors, namely, (i) 10:00 – 24:00 LT, and (ii) 0:00 – 10:00 LT. Sector (i) which is day time and before mid-night includes the time when daily and secondary TEC peaks are expected, while (ii) which is mostly at night is when TEC varies slowly. When slow variation of TEC was expected as in sector (ii) and there were at least a few (> 2) TEC data available, smoothing spline (De-Boor, 1978) data fitting method was used to estimate missing TEC values. In cases where rapid TEC variations are expected as in sector (i) and at least half of the total expected number of data points were filled with TEC data, piece-wise cubic interpolation (De-Boor, 1978) data fitting method was used to estimate missing TEC values. For example, when there were at least 4 measurements in sector (ii) the missing values were obtained by evaluating the fitted function through the existing TEC data values. On the other hand, when there were at least 7 (half the number of hours during 10:00 – 24 LT) TEC values in sector (i), the missing values were obtained by evaluating the fitted function to the available data values. After estimating the missing TEC data from the two sections of the diurnal TEC, the

1      entire diurnal TEC data over a particular grid cell was then considered to

2      estimate the missing values. When there were at least 12 (half the number of

3      hours in a day) values, the missing values were obtained by evaluating a

smoothing spline function fitted to the existing data values.

2. At a particular latitude and local time, the values of TEC along all the longitudes

were divided into western (-20 – 20º E) and eastern (20 – 60º E) longitude

sectors. Each of the longitude sectors contained 8 bins. At night, when there

were at least 3 TEC values over any longitude sector, the missing values were

obtained by evaluating smoothing spline function fitted to the available data

points, while during the day, when there were at least 4 Tec values, the missing

values were obtained by evaluating a smoothing spline function fitted to the

available data points. After estimating the missing TEC values over the two

longitude sectors, the TEC over all longitudes were then considered to estimate

the missing values. At night, when there were at least 8 values, the remaining

values where obtained by evaluating a smoothing spline fitted to the available

TEC data points. The missing values during day-time were estimated when there

were at least 10 measurements available.

3. Procedure 3 is similar to 2, except for variations of TEC as a function of latitude

were considered at specific values of longitude and time. TEC values over the

latitudes were divided into lower (-35 – 0º S) and upper (0 – 35º N) latitudinal

sectors. There were 12 bins in each of the latitudinal sector. To estimate missing

TEC values at night over a latitudinal sector, at least 4 measurements were

required to be available, while during the day, at least 6 values were required.

When TEC data over the combined latitudinal sectors were considered to

estimate the missing values, at least 12 values were required to be available.

After repeating procedures 1 – 3 three times, all the 331,776 bins were filled with TEC

data. For purposes of minimizing the effects of outliers, the diurnal TEC at spatial grid

cells were then separately fitted with smoothing splines which were evaluated to obtain

the TEC data that were later used to determine the model coefficients as explained in

section 3. In order to demonstrate the appropriateness of our estimation of missing TEC

data values and its use for determining model coefficients, we present Figure 1. Panels

(a) – (c) of the figure present the available TEC data (*) and estimated (red line) TEC

values during low, medium, and high solar flux levels, respectively.

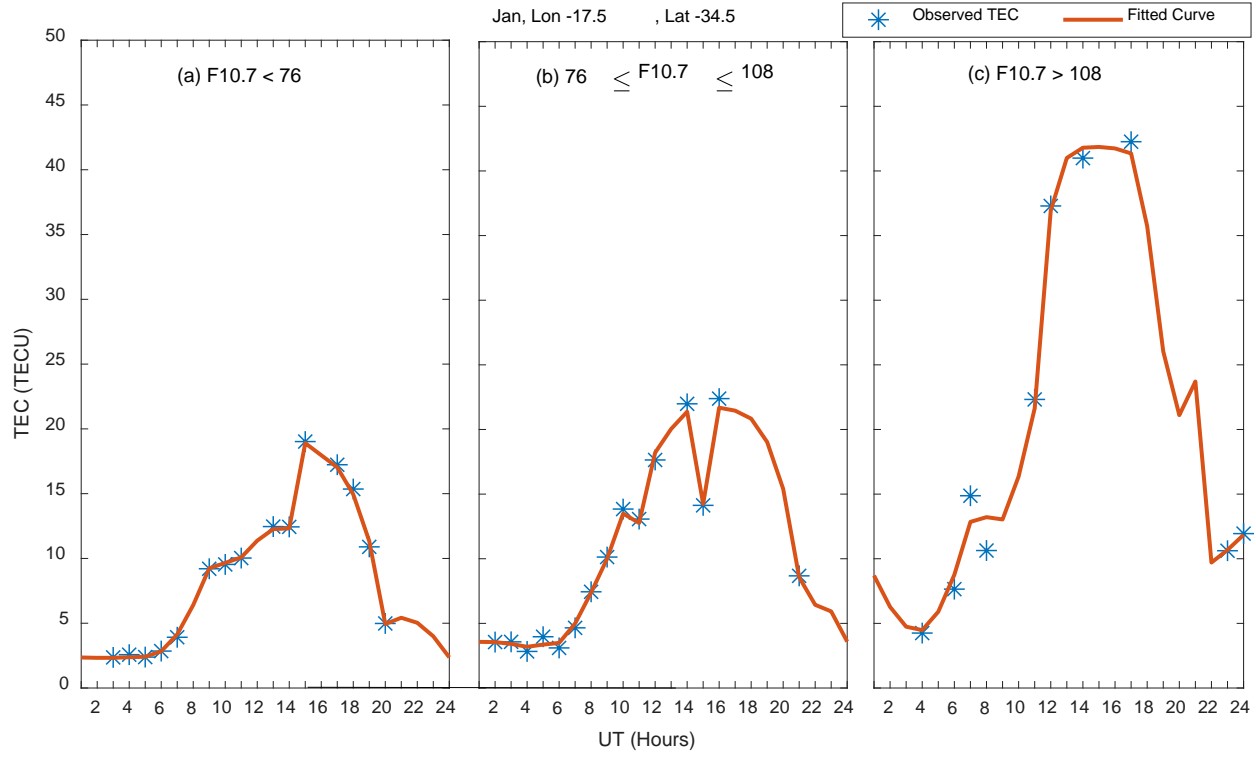

Figure 1: Panels (a) – (c) present available (*) and estimated (red line) TEC values
during low, medium and high solar flux levels, respectively. The data are for the month
of January and fall within the grid cell centered at longitude and latitude of 17.5º W and
34.5º S, respectively.

The TEC data plotted in Figure 1 correspond to January and the grid cell centered at

longitude 17.5º W and latitude 34.5º S. Figure 1 clearly shows that the available and

estimated TEC variations depict the well-known diurnal and solar activity level

dependence patterns. Moreover, the figure shows that the available data values are in

most cases close to the estimated TEC values. Therefore, the estimated TEC values

were then used to obtain the model coefficients.

**3. The Model**

The TEC over the African region was expressed as

$$TEC(t,d,F,\lambda,\phi) = \sum_{i=1}^{24}\sum_{j=1}^{12}\sum_{k=1}^{3}\sum_{l=1}^{16}\sum_{m=1}^{24} a_{ijklm} \times N_i(t) \times N_j(d) \times N_k(F) \times N_l(\lambda) \times N_m(\phi) \qquad (1)$$

where the linear model coefficients $a_{ijklm}$ were determined by the least square fitting

procedure to the 331,776 TEC data values as in Abdu et al. (2003); Jakowski et al.

(2011b); Mungufeni et al. (2015). In Equation 1, $N_i(t)$, $N_j(d)$, $N_k(F)$, $N_l(\lambda)$, and $N_m(\phi)$ are

B splines of different orders to represent variations of TEC with local time, seasons,

solar flux level, longitude, and latitude respectively. Most of the B splines were of

order 2, except for those used to represent LT and latitudinal variations which were of

order 4. The order of splines used to represent LT and latitude was higher to cater for

the rapid variations of TEC with these two parameters. Twenty-four local time nodes 1,

2, ..., 24 were used. For simple interpolation between months, seasonal/monthly

nodes were placed at the 15th day of each month. Solar flux nodes used in the various

14 months are as shown in Table 2. The longitudinal nodes were separated by 5º and

15 placed at longitudes -17.5, 12.5 7.5, ..., 57.5 degrees, while the latitudinal nodes were

16 separated by 3º and placed at latitudes -34.5, -31.5, -28.5, … , 34.5 degrees.

**4. Comparison of Observed and Modeled TEC**

In order to assess the ability of the model to describe the data used to construct it,

modelled data were compared with the binned data that were used to solve equation 1.

The results of the self-consistency check are presented in Figure 2. It is important to

note that validation using data that was not included during modeling is provided in

section 5. Panels in column (i) of Figure 2 present the observed binned TEC data while

column (ii) presents the corresponding modeled TEC data. In column (iii), we present

the differences between the observed and modeled TEC data, referred to as errors. In

Figure 2, rows (a), (b), and (c) correspond to LSA, MSA, and HSA, respectively. The

horizontal magenta lines in Figure 2 and later also in Figure 3 indicate the location of

~0º dip latitude on the corresponding panel. As expected, Figure 2 clearly shows that

the corresponding modeled TEC almost perfectly matches the observed binned TEC. This can be confirmed by the small (< 0.1 TECU) error values presented in panels of column (iii). The variations of the ionosphere with local time, solar flux level as well as location that are exhibited in Figure 2 gives the confidence of relying on the binned data as a good representation of the ionosphere. The physical explanations for these variations are as follows. The increase of both observed and modeled TEC that occurs when solar flux level increases is usually attributed to increased ionizing radiations in X-ray and Extreme Ultra-Violet (EUV) bands, which in turn leads to increased TEC in the ionosphere (Hargreaves, 1992).

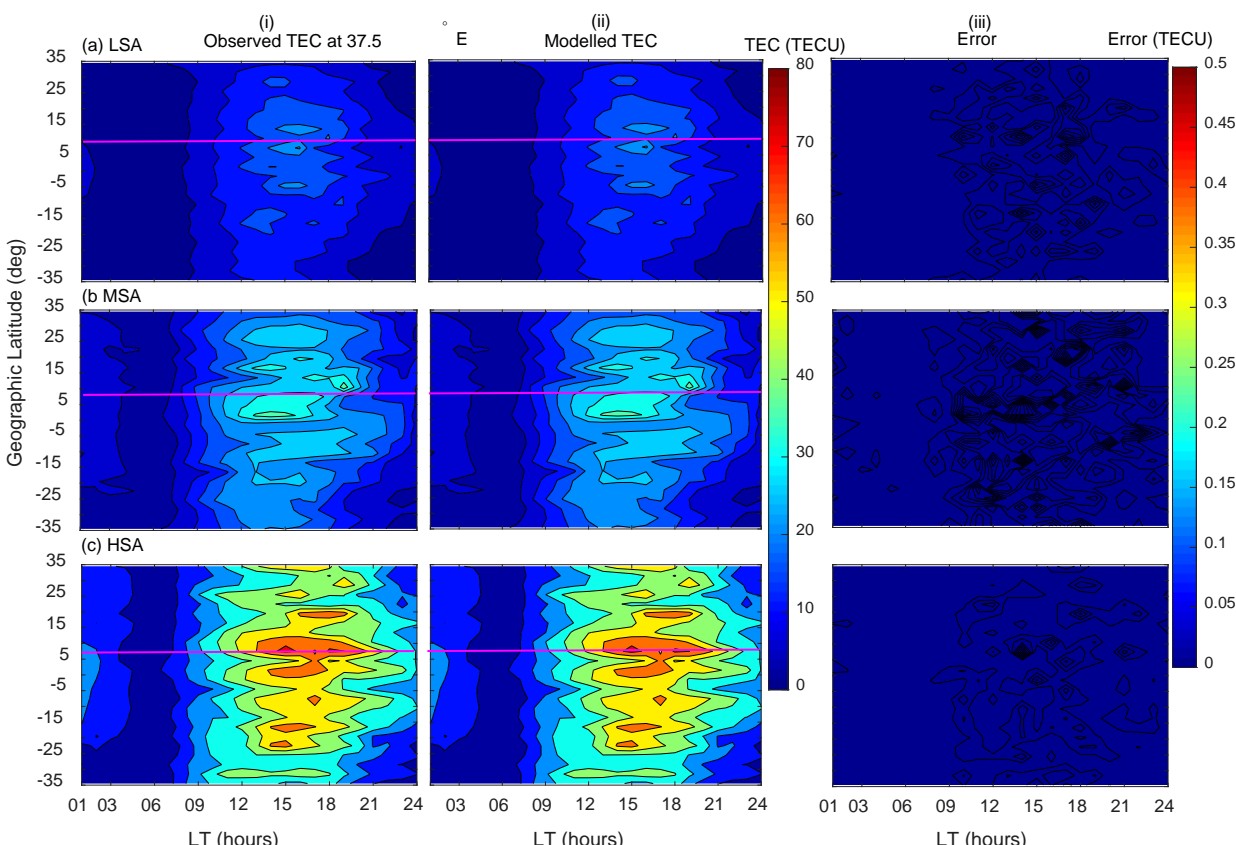

Figure 2. Variation of TEC as a function of geographic latitude and local time in March equinox at 37.5° E. Panels in rows (a) - (c) correspond to LSA, MSA, and HSA, respectively, while panels in columns (i) - (iii) correspond to observed binned, modeled TEC, and difference between observed and modeled TEC (errors), respectively. Magenta line indicates ~0° dip latitude.

The diurnal variation of TEC matches very well with the variation of photo-ionising radiations. At sunrise, the electron density begins to increase rapidly owing to photo-ionization (Schunk and Nagy, 2009). After this initial increase at sunrise, electron density displays a slow rise throughout the day, and then it decays at sunset as the photo-ionization source disappears. Another diurnal feature of variation of TEC exhibited in Figure 2 is the existence of a secondary maximum of TEC. This can clearly be seen in panels of row (c) along the magenta lines, where the first peak occurs at ~15:00 LT and the second at ~18:00 LT. The formation of a secondary maximum of TEC that was mentioned previously may be explained as follows. During the day, the thermospheric wind generates a dynamo electric field in the lower ionosphere that is eastward (Schunk and Nagy, 2009). The eastward electric field, E in combination with the northward geomagnetic field, B produces an upward E$x$B drift of the F region plasma. As the ionosphere co-rotates with the Earth toward dusk, the zonal (eastward) component of the neutral wind increases. The increased eastward wind component, in combination with the sharp day-night conductivity gradient across the terminator leads to the pre-reversal enhancement in the eastward electric field (Batista et al., 1986; Schunk and Nagy, 2009). The F layer therefore rises as the ionosphere co-rotates into darkness. Although in the absence of sunlight after sunset, the lower ionosphere rapidly decays, there exists high electron density at high altitudes, yielding the secondary maximum in TEC.

Panels in rows (b) and (c) of Figure 2 demonstrate the existence of the EIA region, where there exist two belts of high electron density on both sides of $0^o$ dip latitude. The EIA is usually attributed to the upward *ExB* drift which lifts plasma to higher altitudes. The plasma then diffuses north and south along magnetic field lines. Due to gravity and pressure gradient forces, there is also a downward diffusion of plasma. The net effect is the formation of the EIA region (Appleton, 1946). Another feature of EIA that can be seen on panels in rows (b) and (c) of Figure 2 is the asymmetry of the crests. Along $120^o$ longitude sector Zhang et al. (2009) reported the asymmetry of EIA crests. As

described later at the end of this section, the direction of neutral meridional winds in March may favour high values of electron density over the southern crest.

Generally, Figure 2 shows that, the locations outside the EIA region have lower TEC values compared to locations around and within the EIA region. The low values of TEC over locations outside the EIA region might be due to lower elevation angle of solar radiation flux which is responsible for creation of electrons (Schunk and Nagy, 2009). The solar radiation flux is usually low for locations far from the sub-solar point. The latter situation is dominant over locations outside the EIA region, especially in March. The closeness of the sub-solar point to the locations within the EIA regions result into high solar radiations over these locations. As a result, high TEC values were observed over locations within the EIA region.

To demonstrate that the modeled TEC captures TEC variation with seasons, we present Figure 3. In the figure, columns (i) and   (ii)   present   observed   binned   and   the corresponding modeled TEC respectively. Moreover, rows (a) - (d) present TEC data during March, June, September and December, respectively.

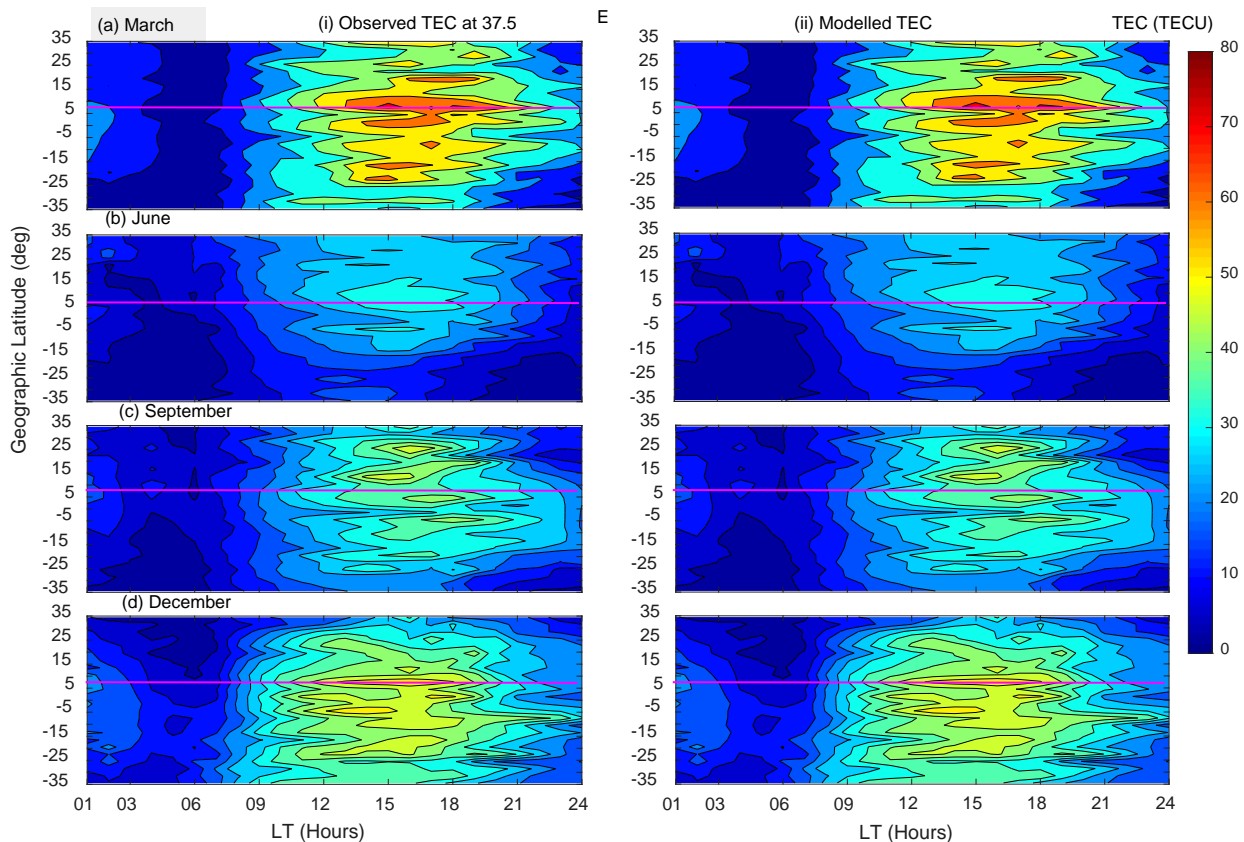

Figure 3. Variation of TEC as a function of latitude and local time in HSA at 37.5°E. Panels in rows (a) - (d) are for March equinox, June solstice, September equinox, and December solstice respectively, while panels in columns (i) and (ii) are observed binned and modeled TEC respectively. Magenta line indicates 0° dip latitude.

As already observed in Figure 2, it can clearly be seen from Figure 3 that the modeled TEC almost perfectly matches the observed TEC data. Among the many features of TEC exhibited by both observed and modeled TEC data, we would like to emphasize the (i) equinoxial asymmetry of TEC, (ii) occurrence of lowest TEC in June solstice, and (iii) high values of TEC in December. Features (ii) and (iii) were recently reported based on a similar data by Mungufeni et al. (2019). The reader may refer to this study for more discussions. Mungufeni et al. (2016a) observed equinoxial asymmetry when studying ionospheric irregularities over the African low latitude region. They observed over the East African region that, the irregularity strength in March equinox was higher than that in September equinox. They attributed the equinoxial asymmetry to meridional winds in

March which might blow northward. Such a direction would lift plasma up where recombination is not common. On the other hand, in September, the winds might blow southward. This could lead to recombination at low altitudes.

## 5. Model Validation

### 5.1 Validation using reserved COSMIC RO TEC

In addition to comparing observed binned TEC with the corresponding modeled TEC, we validated our model using observed TEC in the years 2012 and 2018. The data during these two years were not used in developing the model. The TEC data in the years 2012 and 2018 were binned according to local time and spatially in a similar manner to that mentioned in subsection 2.2. The corresponding local time, day of the year, solar flux, and spatial coordinates of the data were noted and then used to generate the corresponding modeled TEC. Despite the advantages of B spline modeling mentioned in section 1, one of its limitations is the inability to extrapolate. Therefore, in situations where the solar flux level is higher (lower) than those specified in Table 2, the maximum (minimum) value in the table was used to generate the corresponding modeled TEC. This idea was also applied when the day number of year, longitude, and latitude values were higher (lower) than those specified in section 3.

Figure 4 presents a scatter plot showing the observed TEC against the corresponding modeled TEC. The red line in the figure indicates linear least squares fit to the data in the panel. Furthermore, indicated in Figure 4 are: (i) the correlation coefficients, r, (ii) the r squared values, (iii) the number of data points, n plotted and (iv) the root mean squared error, RMSE when the modeled TEC is used to represent the observed TEC.

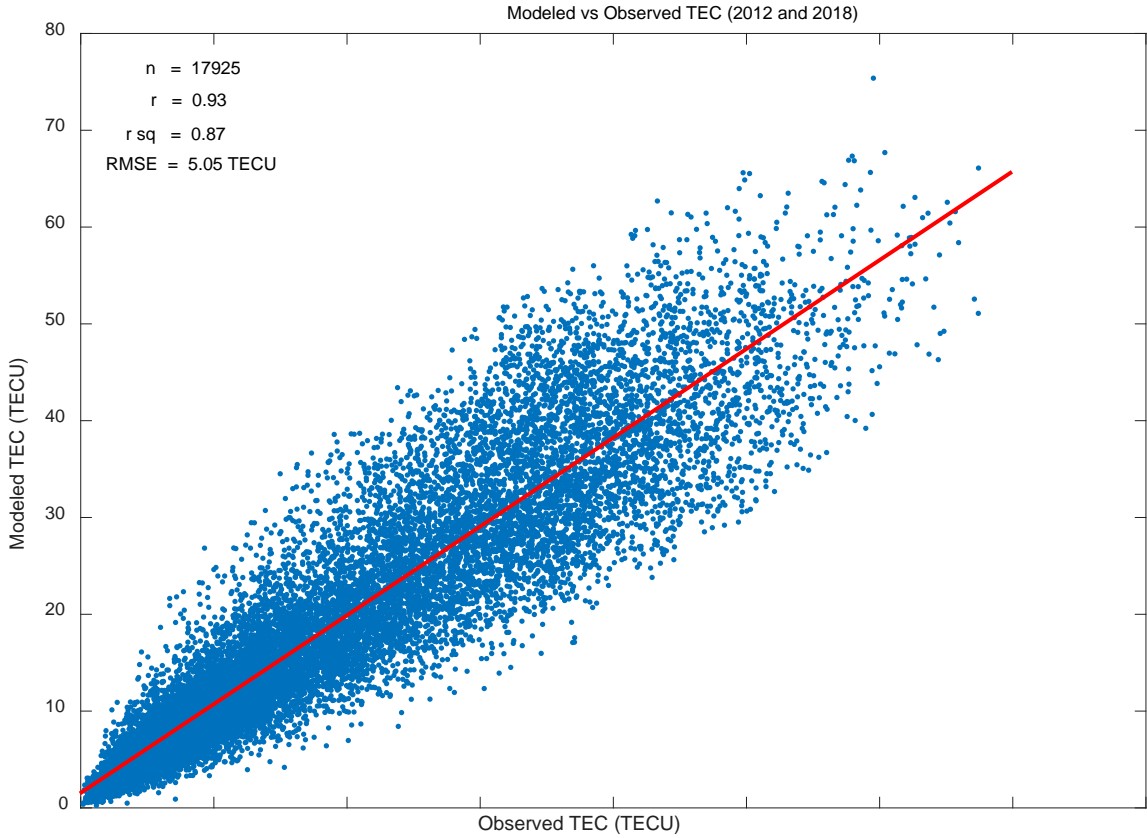

2  Figure 4. Scatter plot of observed TEC against modeled TEC.

The following observations can be noted from Figure 4. (i) The modeled TEC correlates

highly (r ~0.93) with the observed TEC. (ii) The r squared values indicate that high

proportions (~87 %) of the variations in the observed TEC can be predicted

by the modeled TEC. (iii) The RMSE value of 5.05 TECU signify that the modeled TEC

closely approximates the observed TEC.

In order to show that the observed and modeled TEC have similar magnitudes in

addition to their similar variation depicted in Figure 4, we computed the differences

between corresponding values of the data plotted in the figure. These were referred to

as errors. We also computed the percentage of the different errors. The left and right

vertical axes in Figure 5 present the distribution of the number of observed errors and

their percentages, respectively. It can be seen from the figure, the errors are randomly

distributed since the distribution curve is symmetric about 0 TECU. Indeed, the magnitudes of the modeled TEC values are close to that of the observed TEC since the majority of the error values are close to zero.

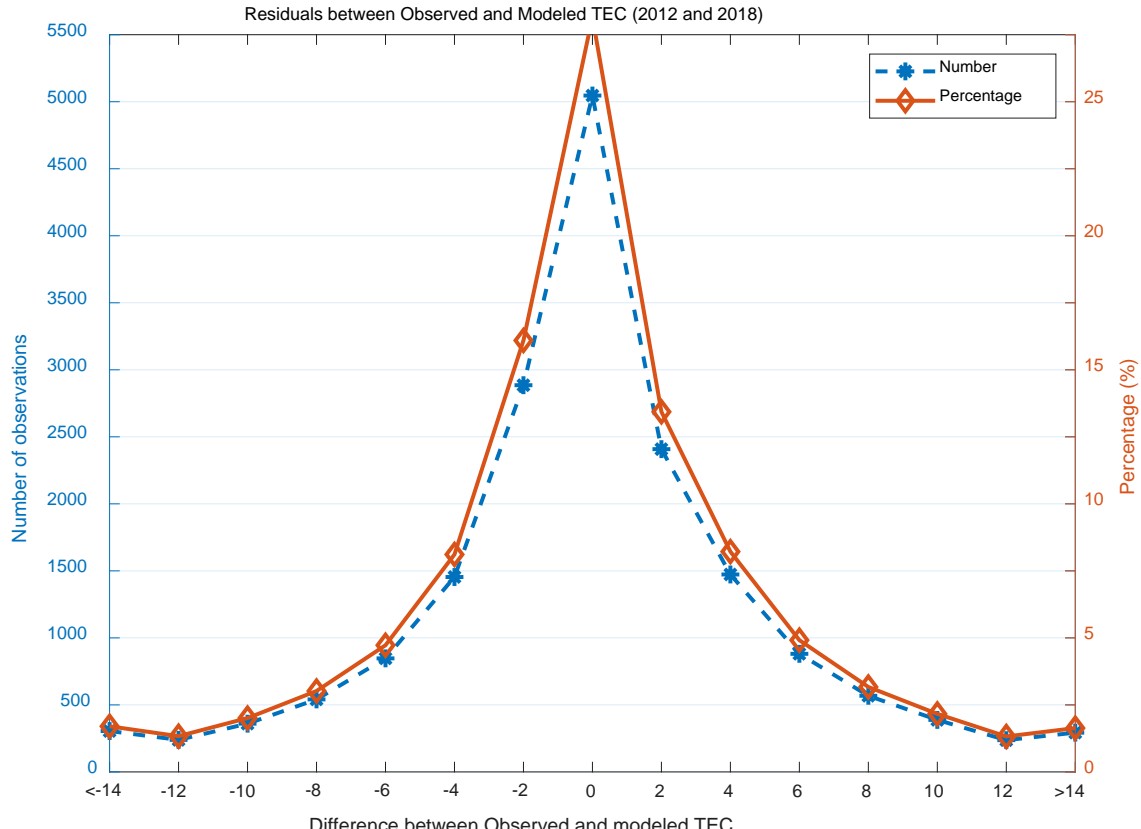

Figure 5**.** The blue and red curves show the distribution of the number of observed errors (difference between observed and modeled TEC) and the percentage of the errors, respectively.

The cases of high error values (> 10 TECU mostly have < 2.5 % occurrence probability, as can be seen on the right vertical axis. These high errors may be partly attributed to the limitation of spline modeling technique (inability to extrapolate) which was discussed earlier in this subsection 5.1.

**5.2 Validation using ionosonde TEC measurements**

The TEC data measured by the digisonde ionosonde stations over South Africa located at Hermanus, Grahamstown and Louisvale can be accessed from the National Oceanic and Atmospheric Administration (NOAA) website via the link, ftp://ftp.ngdc.noaa.gov. The data obtained from the NOAA website is in form of auto-scaled ionospheric parameters such as peak height in F2 layer, critical frequency in F2 layer, and TEC which are stored in Standard Archiving Output (SAO) format files. It should be noted that the TEC data provided in SAO files are obtained by integrating electron density profiles up to altitude of ~700 km. More details about the auto-scaling program (real-time ionogram scaler with true height (ARTIST)) and the electron density profiles they produce can be found in Reinisch and Huang, 2001 and Klipp et al., 2020.

Figure 6 presents with magenta lines the diurnal patterns of TEC measured by ionosonde stations at Hermanus (panels in column (i)), Grahamstown (panels in column (ii)) and Louisvale (panels in column (iii)). The corresponding TEC generated by our spline technique model (spline), Nequick 2, and IRI-2016 are superimposed with red, green and blue lines, respectively. We need to mention that during computation of TEC using NeQuick 2 and IRI-2016, the height was limited to the approximate altitude of the COSMIC satellites (800 km). Moreover, for the case of IRI-2016, NeQuick model option was specified to estimate topside electron density.

The panels in rows (a) - (c) show TEC on day of year 170 (June), 260 (September), and 350 (December), respectively. All these three days of the year 2013 were geomagnetically quiet. Preliminarily, Figure 6 appears to reveal that IRI-2016 either overestimates (December) or underestimates (June and September) the TEC measured by the ionosonde stations. On the other hand, our spline model and NeQuick 2 seem to depict good correspondence between the observed and the modeled TEC. It can also be seen from Figure 6 that over a particular station, the shape of curves on different days representing TEC generated by the IRI-2016 and NeQuick 2 models are similar. This is expected since these two models were meant to reproduce monthly median values of the ionosphere. This means that our model, based on spline functions may capture better the day-to-day variability of the ionosphere.

Figure 6: Magenta color shows diurnal TEC observed by ionosonde stations at Hermanus (panels in column (i)), Grahamstown (Panels in column (ii)), and Louisvale (Panels in column (iii)). The green, blue, and red colors show TEC estimations using NeQuick 2, IRI-2016 and Spline models, respectively. Panels in rows (a) - (c) show diurnal TEC during the year 2013 on DOY 170, 260, and 350, respectively.

We generated such data plotted in Figure 6 for geomagnetically quiet days of the entire year 2013 and then performed statistical analysis of the observed and the model TEC data. Table 3 presents in columns 3 the correlation coefficients, r for the correlations between modeled and ionosonde TEC. Moreover, the table presents the RMSE when the ionosonde TEC was estimated using the models listed in column 2. The number of observations, n over each station that were used to determine, r and RMSE are put in brackets below the station name.

Table 3: Correlation coefficients, r and RMSE associated with estimation of TEC observed by ionosonde stations using models

| Ionosonde Station /number of observations | Model | R | RMSE (TECU) |
|---|---|---|---|
| Hermanus (n = 5,110) | Spline | 0.92 | 4.64 |
| | IRI-2016 | 0.86 | 5.45 |
| | NeQuick 2 | 0.92 | 4.10 |
| Grahamstown (n = 4,450) | Spline | 0.88 | 5.56 |
| | IRI-2016 | 0.82 | 6.29 |
| | NeQuick 2 | 0.86 | 5.27 |
| Louisville (n = 4,543) | Spline | 0.94 | 3.82 |
| | IRI-2016 | 0.87 | 5.62 |
| | NeQuick 2 | 0.94 | 3.73 |

It can be seen from Table 3 that the r values associated with NeQuick 2 and spline based model are consistently better when compared with that of IRI-2016. Moreover, the RMSE values associated with IRI-2016 are the highest in all the cases. These two observations indicate that compared to spline and NeQuick 2, IRI-2016 poorly estimates TEC at the locations of the ionosondes. The RMSE values associated with NeQuick 2 are always slightly lower than that of spline, while the r values associated with spline are mostly comparable or slightly higher than that of NeQuick 2. These discussions demonstrate that our spline model generates TEC values consistently with that observed by ionosondes. This implies that equivalent TEC measured by ionosondes over mid-latitude locations which do not have ionosonde stations can be predicted fairly well using our model. We might validate our model over low-latitude region that falls

within the current study area when in future ionosonde observations become available over the region.

## 5.3 Comparison of our model with existing regional models

It would be good to compare error levels produced when some measured TEC are compared with modeled TEC generated by (i) the existing regional TEC models discussed in section 1 and (ii) our spline technique TEC model. We may not perform such analysis since models in (i) are based on electron density integrated from ground up to GPS satellites (~20,200 km), while model in (ii) is based on electron density integrated up to ~800 km. However, we present Figures 7 and 8 to compare EIA features captured by our spline technique model with those by the neural networks technique of Okoh et al., (2019). The TEC plots based on the neural networks technique can be obtained from MATLAB Central website (Okoh et al., 2019). (https://www.mathworks.com/matlabcentral/fileexchange/69257-african-gnss-tec-afritec-model?s_tid=prof_contriblnk). We present in Figure 7 examples of TEC generated by neural network model during the year 2012 at 11:00 UT. Over the East African sector (LT = UT + 3), this time translates to 14:00 LT and falls within the range of LT when EIA exists over the region (Mungufeni et al., 2018). Panels (a) and (b) in Figure 7 present TEC during March (DOY 81) and September (DOY 260) equinoxes, respectively, while (c) and (d) present during June (DOY 171) and December (DOY 347) solstices, respectively. It is important to mention that these 4 days were geomagnetically quiet.

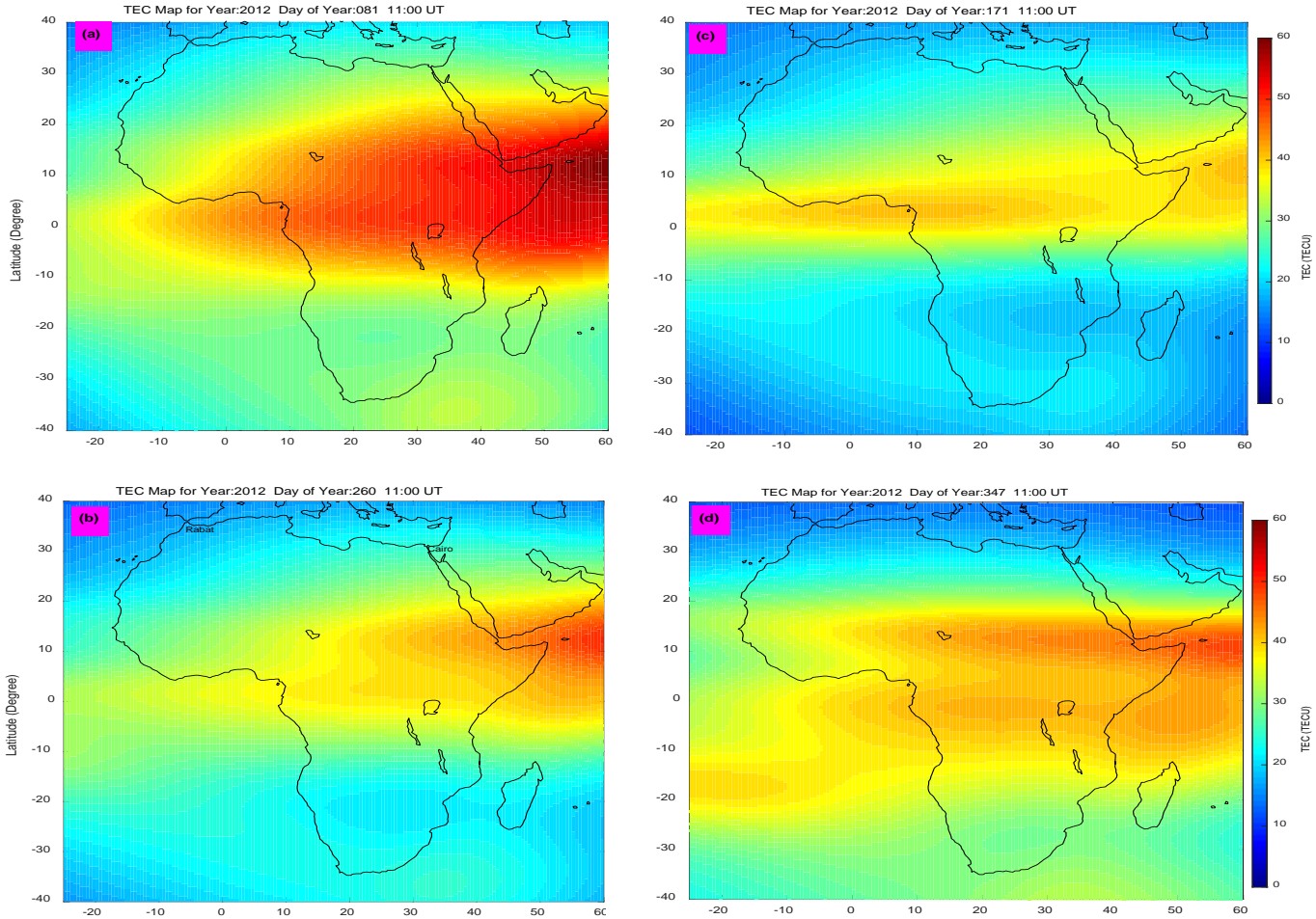

Figure 7: Neural Network TEC maps during the year 2012 at 11:00 UT. Panels (a) and

(b) are for March (DOY 81) and September (DOY 260) equinoxes, respectively, while

(c) and (d) are for June (DOY 171) and December (DOY 347) solstices, respectively.

In order to generate TEC maps using our model for purposes of comparing with TEC

maps in Figure 7, we noted and used the F10.7 flux values on the days indicated in the

figure. The TEC maps generated using our model that correspond to TEC maps

presented in Figure 7 are presented in Figure 8.

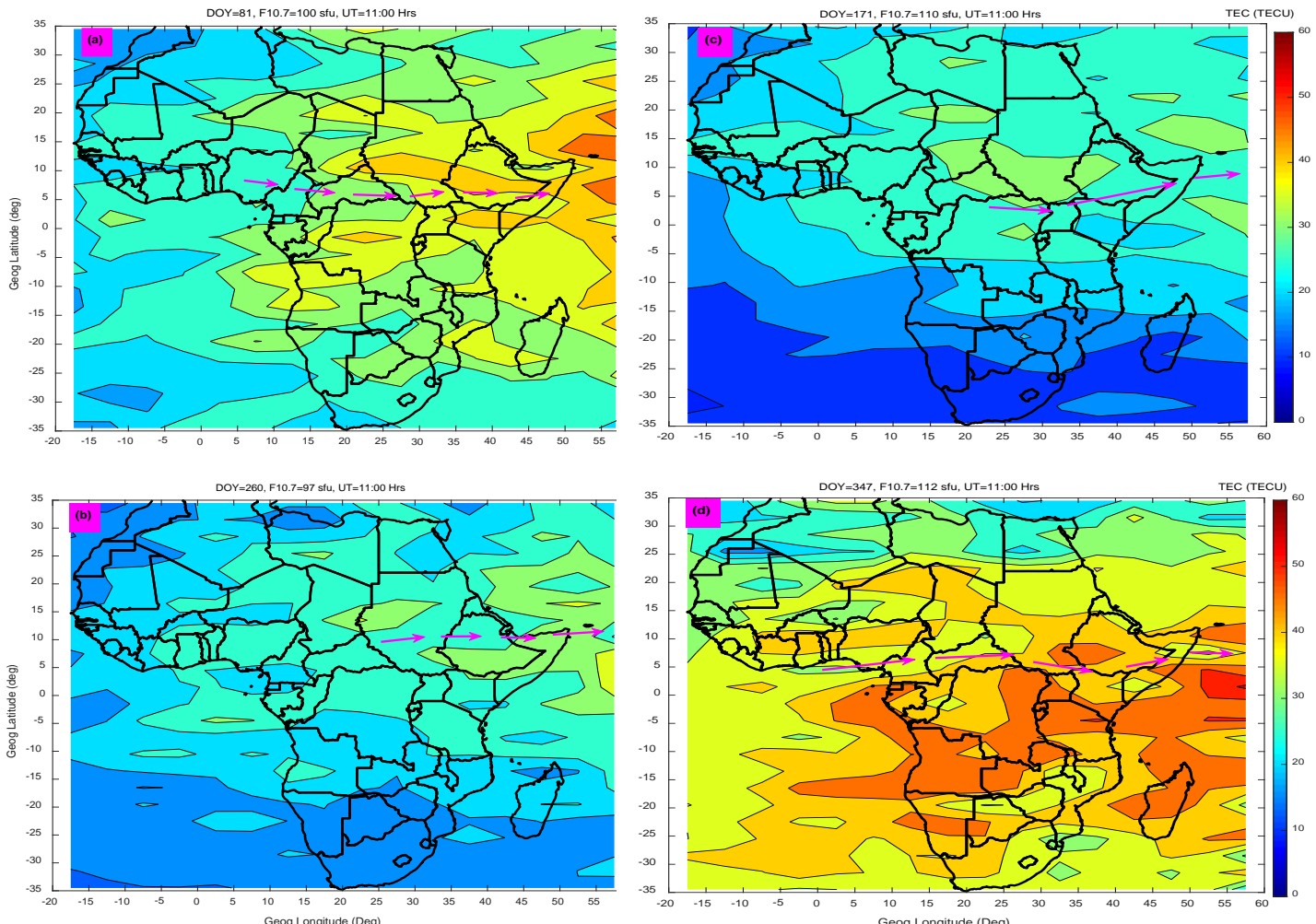

Figure 8: Similar to Figure 7, but generated by spline modeling technique. Magenta arrows indicate approximate locations of EIA trough.

Unlike our TEC maps in Figure 8 which clearly show the EIA trough (see magenta arrows) in all the seasons, the neural network technique TEC maps (Okoh et al., 2019) of Figure 7 only clearly capture the EIA trough in December solstice. As pointed before, this short fall in neural network TEC model might be due to poor amount of data to represent day of year during model development. Another observation that can be made from Figures 7 and 8 is that unlike the neural network model which yields smooth spatial TEC variation, the spline modeling technique does not yield smooth spatial TEC variation. In real life, measurement or observed values rarely vary smoothly. Since the

spline modeling technique produces results (see Figure 2) which demonstrate that the modeled data matches almost perfectly the observed data, it is expected that the spatial variations of TEC in maps of Figure 8 are not smooth.

## 6. Conclusions

This study developed a model of TEC measured by COSMIC satellites. The TEC data were binned according to local time, seasons, solar flux level and spatially. The coefficients of B splines that were fitted to the binned data were determined by means of the least square procedure. As expected, the modeled TEC almost perfectly matched the corresponding observed binned TEC data. The model was validated with independent data that were not used in the model development. The validation revealed that (i) the observed and the modeled TEC correlate highly ($r = 0.93$), (ii) the coefficient of determination $R^2$ which is the proportion of variance in the observed data predicted by our model was 87 %, and (iii) the modeled TEC closely approximates the observed TEC (RMSE of 5.05 TECU). Due to the extensive input data and the applied modeling technique, we were able to reproduce the well known features of TEC variation over the African region. Further validation of our model using TEC obtained from ionosonde stations over South Africa at Hermanus, Grahamstown and Louisville reported r values > 0.92 and RMSE < 5.56 TECU. These validation results imply that our model can estimate fairly well TEC that would be measured by ionosondes over locations which do not have the instrument.

**Acknowledgments**

This study received financial support from research number, 018-1370-20 in the department of Astronomy and Space Science of Chungnam National University which was awarded by the Air Force Research Laboratory of the United States of America. The first author, Patrick Mungufeni greatly appreciates the immense contribution of Prof.

Claudia Stolle towards shaping the presentation of the manuscript. We thank the developers of the IRI and NeQuick models for making their models available. Dst data is provided by the World Data Center for Geomagnetism at Kyoto (http://swdcwww.kugi.kyoto-u.ac.jp/). Kp data isprovided by GFZ Potsdam, ftp://ftp.gfz-potsdam.de/pub/home/obs/kp-ap/. F10.7 flux data was obtained from http://www.swpc.noaa.gov/, while ionPrf files used to derive COSMIC TEC were obtained from http://cosmic-io.cosmic.ucar.edu/cdaac/index.html. We thank NOAA for availing ionosonde data via the link, ftp://ftp.ngdc.noaa.gov.

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
