# Peer review of "Modeling Total Electron Content derived from radio occultation measurements by COSMIC satellites over the African region"

_Annales Geophysicae, 2019_

## Referee Comment (RC1) · Anonymous Referee #1 · 24 Dec 2019

General comments This paper examines the possibility to Total Electron Content (TEC) over the African region TEC data derived from radio occultation measurements done by theConstellation Observing System for Meteorology, Ionosphere, and Climate (COSMIC) satellites for Geomagnetically quiet time (Kp < 3 and Dst > -20 nT) data during the years 2008 - 2011, and 2013 – 2017.

Specific comments 1) An author of this paper (Patrick Mungufeni) along with a long list of other authors have recently published the following paper Okoh, D., Seemala, G., Rabiu, B.,Habarulema, J. B., Jin, S., Shiokawa, K., et al. (2019). A neural network‐based ionospheric model over Africa from Constellation Observ-

ing System for Meteorology, Ionosphere, and Climate and Ground Global Positioning System observations. Journal of Geophysical Research: Space Physics, 124. https://doi.org/10.1029/2019JA027065

In that particular paper the authors perform an adjustment using Neural Networks according to which they correct the reasonable discrepancy between TEC from ground based receivers (up to 22000 Km) and occultation measurements (up to 700 Km. They seem to apply no such procedure in this paper. This is a major problem of this paper. They also need to make special reference to that paper. 2) Maybe they should compare the output of the NN model out of that paper with the output of the spline model for this paper despite that the COSMIC dataset is used as a basis for both models. In this way they will prove their approach for this paper (omitting any correction for the plasmaspheric contribution which is expected to be high at middle African latitudes). 3) The authors do not provide any scheme by which they would reject any unrealistic COSMIC profiles. There have been numerous validation studies with Digisondes that verify this problem especially in the bottomside. 4) I strongly suggest to compare the output of their model with ionospheric TEC (up to 700 km) from all over four stations Digisonde stations over South Africa https://spaceweather.sansa.org.za/products-and-services/current-conditions/ionograms . This will provide a much more realistic comparison test to their model.

---

## Short Comment (SC1) · 1 Feb 2020

Responses to interactive comments on**, "Modeling Total Electron Content derived from radio occultation measurements by COSMIC satellites over the African Region"**

**By Mungufeni et al.**

January 24, 2020

We thank the anonymous referee for the comments.

**Comment:**

An author of this paper (Patrick Mungufeni) along with a long list of other authors have recently published the following paper: Okoh, et al. (2019). A neural network based ionospheric model over Africa from COSMIC and Ground GPS observations. Journal of Geophysical Research: Space Physics, 124. https://doi.org/10.1029/2019JA027065. In that particular paper the authors perform an adjustment using Neural Networks according to which they correct the reasonable discrepancy between TEC from ground based receivers (up to 22000 Km) and occultation measurements up to 700 Km. They seem to apply no such procedure in this paper. This is a major problem of this paper. They also need to make special reference to that paper.

**Response:**

Indeed, Patrick Mungufeni contributed to the paper in the comment which was published on Thur, Dec 12, 5:23 PM. The current manuscript under discussion was submitted on Saturday, Nov 23, 2:09 PM (Korean time). Therefore, we could not reference Okoh et al, (2019) since it was published later after the current submission. Below are the screen shots of emails to prove the dates. Anyway, we shall reference Okoh et al. (2019).

[Figure]

[Figure]

Although the reviewer is recommending creation of data base consisting of both ground and space based TEC measurements, such data base may be subjected to criticism. For example, the observation in Okoh et al. (2019) where the ratio between ground

based and COSMIC TEC varies spatially implies that neural network may not learn the relationship between the two data sets over locations which only have COSMIC TEC data. We have highlighted with pink boxes in Figure below such regions which mostly have COSMIC TEC. The Figure was taken from Okoh et al. (2019). Over pink boxes, the adjustments made to COSMIC TEC may not be trusted because of large distances over which interpolations are done.

[Figure]

When a study opts to have both adjusted COSMIC TEC data and ground based GPS TEC data, some locations will be represented by adjusted COSMIC TEC (remember not trusted) while others will be represented by ground based GPS TEC. Obviously, there is still disparity. For purposes of consistency, It might be fair to use entirely adjusted

COSMIC TEC since it can also be available where there is ground based GPS TEC. Since we do not trust the current known procedures for adjusting space based observations (Okoh et al. (2019) and Mungufeni et al. (2019), *Estimation of equivalent ground-based total electron content using CHAMP-based GPS observations*, Adv in Space Res 64, 199 - 210) the current manuscript used only COSMIC TEC without any adjustment.

**Comment:**

Maybe they should compare the output of the NN model out of that paper with the output of the spline model for this paper despite that the COSMIC dataset is used as a basis for both models. In this way they will prove their approach for this paper (omitting any correction for the plasmaspheric contribution which is expected to be high at middle African latitudes

**Response:**

The suggestion in the comment will be implemented.

**Comment:**

The authors do not provide any scheme by which they would reject any unrealistic COSMIC profiles. There have been numerous validation studies with Digisondes that verify this problem especially in the bottomside.

**Response**:

Empirical modeling requires adequate data for the mathematical functions to capture the physics inherent in the data. However, to minimize measurement errors, studies that have used COSMIC data commonly reject measurements with horizontal smear > 1500 km. We have presented in Figure below the number of COSMIC TEC measurements per day during the year 2013 over the longitude and latitude ranges of -15 – 60$^o$ and -35 – 35$^o$, respectively.

[Figure]

The blue dots indicate COSMIC TEC measurements when the horizontal smear is < 1500 km, while the red stars indicate COSMIC TEC measurements without limitation of horizontal smear. It can be noticed that when the horizontal smear is limited, ~40 observations may be made per day. Obviously, the 40 measurements may not cover very well all the 24 hours in a day and all the grid cells. This shows clearly that the seasonal or day number of year variation does not have good input data for the entire African region. In order to have fairly adequate data, we did not apply restriction to the horizontal smear. Therefore, in a day, there were about 80 observations as shown with red stars in Figure above.

We established that the COSMIC TEC data values with smear > 1500 km do not introduce alarming errors. This was done by analyzing COSMIC TEC data which were coincident with TEC observed by ionosonde stations at Hermanus, Grahamstown, and Louisvale. The observations of the year 2013 were considered. Table below presents the root mean squared error between (i) ionosonde and COSMIC TEC without limiting

the horizontal smear, and (ii) ionosonde and COSMIC TEC with horizontal smear limited to 1500 km.

| Station | Smear < 1500 km | | No limitation | |
|---|---|---|---|---|
| | Number of observations | RMSE (TECU) | Number of observations | RMSE (TECU) |
| Hermanus | 38 | 1.838 | 65 | 2.256 |
| Grahamstown | 34 | 6.479 | 73 | 7.923 |
| Louisvale | 42 | 2.765 | 91 | 3.252 |

The table shows that the RMSE for the two cases over a particular ionosonde station are not grossly different. Based on these results, trading off accuracy may not be costly compared to trading off adequate need of data. Therefore, we decided not to impose any restriction on the horizontal smear. Although the RMSE appear to be smaller when the smear < 1500 km, some of the data points that were subjected to this restriction are also far from the linear least squares fitting line. See blue stars in Figure below.

[Figure]

Most likely, the ~80 COSMIC TEC data points in a day may not still cover very well all the 24 hours in a day and all the grid cells. This problem might be solved by adopting appropriate data binning criteria. Therefore, instead of binning data according to year, we binned data according to only three different solar flux levels. This technique proved to be good and it was published in Mungufeni et al, (2019), *Characterization of Total Electron Content over African region using Radio Occultationobservations of COSMIC satellites*, Adv in Space Res 65, 19 – 29.

**Comment:**

I strongly suggest to compare the output of their model with ionospheric TEC (up to 700 km) from all over four stations Digisonde stations over South Africa https://spaceweather.sansa.org.za/products-andservices/current-conditions/ionograms. This will provide a much more realistic comparison test to their model

**Response:**

The suggestion in the comment will be implemented.

---

## Referee Comment (RC2) · Anonymous Referee #2 · 2 Feb 2020

Dear Editor,

Please find below the review of the manuscript "Modeling Total Electron Content derived from radio occultation measurements by COSMIC satellites over the African Region" submitted to Annales Geophysicae by Patrick Mungufeni et al.

The manuscript presents an empirical model describing ionosphere total electron content over African region. Authors use experimental TEC data obtained using dual frequency GNSS RO receivers onboard of COSMIC satellites to construct the model. They validate the model using same type of data that was used to construct the model but for a different period.

**General comments:**

General impression is that the present work has no contribution to the current understanding of the low-latitude ionospheric physics/modelling. The work brings a little science and the newly created model could hardly be used in any real-life application. Authors are making too many assumptions and mistakes, sometimes trying to deliberately present performance results better than they are. Moreover, the performance of the model has not been compared to any other well-known model, leaving a room for doubts. Therefore, I recommend the manuscript (in its present form) is **rejected**. At the same time, the work might be improved and worth publication after substantial modifications. Please find below a list of critical issues along with possible improvements/corrections for a potential future re-submission.

**Critical/Major comments:**

P.1 L.27: Replace "good" with "applied". Otherwise, provide a proof of the model "goodness"

P.2. L.35-38: Not all GNSS systems support ionospheric corrections. E.g. GLONASS does not broadcast any ionospheric model parameters. Correct the sentence accordingly.

P.2 L.40: Provide a reference to the original description of Klobuchar model:

*"Klobuchar JA (1987) Ionospheric time-delay algorithm for single frequency GPS users. IEEE Trans Aerosp Electron Syst 23(3):325–331. https://doi.org/10.1109/TAES.1987.310829"*

P.2 L.41-42: NeQuick G model is based on the NeQuick model, but not NeQuick 2. Correct the statement and the reference accordingly, e.g.

*"EC (2016) European GNSS (Galileo) Open Service—Ionospheric correction algorithm for Galileo single frequency users, Issue 1.2, Sept. 2016, European Commission"*

P.2. L.42: Change "The NeQuick is" to "The NeQuick and its subsequent modifications (NeQuick G and NeQuick 2) are"

P.2 L.53: IRI model does not provide information about "electron and ion velocities". It only provides information about equatorial vertical ion drift. Correct the sentence accordingly.

P.2 L.55-56: Change "The model is primarily" to "IRI is an empirical model primarily"

P.3 L.74: Change "GIM" to "global ionosphere model", as GIM is already defined to be Global Ionosphere Map.

P.3 L.76: Change "GIM model" to "global ionosphere model"

P.3 L.80-82: The high values of RMS in low latitude region provided by CODE is, primarily, due to the inability of the selected model function (spherical harmonics) to describe ionospheric structure in low latitude. Modify the sentence accordingly.

P.3 L.84: Change "the GIM model" to "global models"

P.4 L.115: Author use TEC integrated up to COSMIC satellite heights (800 km) to construct the model (*"integration being done up to the altitudes of the COSMIC satellites"*). However, the topside TEC values (according to numerous studies, e.g. by Bilitza 2009, Yizengaw 2008 etc.) can reach from 10% to 80% of the total electron content (from ground to GNSS satellite heights). This fact significantly reduces the scientific value and application of the developed model. Essentially, the model is useless for GNSS applications.

P.5 L.124-126: This statement "*Since the magnitudes of the TEC obtained from COSMIC occultation 124 measurements are close to ground based GNSS TEC*", is not consistent with the previous statement and studies by Mungufeni et al. 2019. Where they show that, depending on the location, the RMS error can vary from 2 to 8 TECU and error distribution plots show values from -24 to 20 TECU. Such large errors cannot be considered "*close to ground-based GNSS TEC*". Authors, at least, are expected to provide information about relative TEC errors (in %, rather than TECU) to claim that errors can be tolerated (if so).

P.6 L.150: The title of the reference Emmert et al. 2010 is incorrect:

*Emmert, J. T., Richmond, A. D., and Drob, D. P.: Statistical analysis of the correlation 412 between the equatorial electrojet and the occurrence of the equatorial ionisation 413 anomaly over the East African sector, J. Geophys. Res., 15; A08322; 414 doi:10.1029/2010JA015326, 2010.*

P.6 L.157-167: The selected spatial resolution of 15$^{\circ}$ in longitude and 5-8$^{\circ}$ in latitude is too coarse to describe the ionosphere reasonably, especially for the low latitude region, where TEC is changing dramatically from the crest down/up to two peaks of EIA. E.g. GIM maps (the source of the data for most of the empirical models discussed by the authors in the introductions section) use at least 5$^{\circ}$ by 2.5$^{\circ}$ resolution (lon and lat). Moreover, 15$^{\circ}$ in longitude corresponds to 1 hour in LT. Gradients in TEC as a function of LT during sunrise and sunset hours may reach tens of TECU per hour (e.g. Mungufeni et al. 2019, Fig. 2). Therefore, such coarse spatial resolution in longitude will lead to big errors in the model description.

P.6 L.170: The whole solar cycle 24 has relatively low solar activity level compared to the two previous ones. Nevertheless, even if we look only at the 24$^{th}$ solar cycle, 2011 and 2016 could hardly

be attributed as years of high solar activity level. Please, modify the statement accordingly (e.g. as it is done on P.7 L.182).

P.7 L.189: Please clarify, how 36 solar flux bins were obtained. From the description, it is only 3 solar flux ranges and 12 months, that gives 36 (3x12). But when listing by a variable, only number 3 has to be specified, as it is done, for example with the rest of the variable (hour, lat and lon). Indeed, if we take 60,480 TEC values indicated in L.189, this number can be obtained by multiplying 5x14x3x12x24, but not 5x14x36x12x24.

P.8 L.205: According to the definition of cubic spline, it is a spline constructed of piecewise third-order polynomials, meaning none of the B splines used in the model were cubic (order 2 and 4). Change the "cubic B spline" into "B spline of different orders" throughout the text and abstract.

P.9 L.218-220: Consider changing this sentence to something like "In order to assess the ability of the model to describe the data used to construct the model, modelled data were compared to the experimental one. The results of the self-consistency check are presented in Figure 1."

P.9 L.228-229: It is surprising that the authors compare the results of the climatological model (i.e. model where input data were averaged over time, e.g. one month) with GIM map for a single day of that month. Such a comparison is not correct. On top of that, by looking at TEC maps obtained from COSMIC and later by B spline model (columns 2 and 1), one can hardly see any separation between the peaks of the EIA, that can, taking into account averaging in all the bins (e.g. lat and lon) performed by authors, hardly be comprehended.

P.9 L.231-232: By looking at the color plots, a reader can hardly assess the performance of the model. It is suggested, in addition to the plots, to present/discuss the results of the mismodelling in terms of a bias and RMS of the error.

P.10 L.250-252: From Fig 1 it can not be clearly understood the secondary maximum if any, especially at -20 lat. Please, if you discuss a feature, try demonstrating it clearly to the reader. A separate figure, or at least, a dashed line at -20 and 4 in Fig 1 is needed to support the statement.

P.11 L.269-270: In row (b), Fig 1, none of the panel show peaks of the EIA. There is no clear separation of the crest and peaks of EIA. Nor in panels b1/b2 neither in b3. Modify the sentence accordingly.

P.277-279: The structure of the crest might differ based on various factors (including level of the geomagnetic disturbance). However, when taken as an average, a clear 2 peak structure is present in low latitudes, representing EIA.

P.12 L.298-299: The science question in this case is not how to model the observed data, but how to explain the data. What is the physical explanation for the absence of the EIA structure (two peaks and the crest) in TEC values calculated from the ground up to COSMIC satellite heights (~800km).

And whether this phenomena is not a limitation of the technique applied to calculate TEC. Namely, TEC computed by integrating electron density profile, that by itself is a product of RO inversion, is subject to big errors, especially in places where big horizontal gradients exist (read, e.g. M.M Shaikh et al., Implementation of Ionospheric Asymmetry Index in TRANSMIT Prototype, DOI: 10.5772/58551). Without understanding the reasons of the observed behavior all the modelling efforts are meaningless.

P.13 L.313: One cannot see the "perfect match" of the observed and modelled data just by looking at the plots. At least a third row in form of difference map (error map) has to be presented to visually assess the error level. Moreover, statistical results (e.g. RMS and bias of the error) must be presented in order to make such a bold conclusion.

P.13 L.312-324: Authors do not discuss at all the TEC behavior observed in September at lat ~ -20, where its diurnal variation has a maximum during local night hours (21-03 LT). This maximum seems to exceed any other TEC values on this plot (row c, column 1 and 2) and looks like an error in the data processing. Such behavior seems to have no physical explanation.

P.14 Section 5: The authors fail to explain why they need yet another TEC model. Unless the performance of the newly created model is compared to existing models and it is demonstrated that its any better than the rest of the models present on the "ionosphere model market" (e.g. IRI, NeQuick, NTCM etc.), there is very little value in the study (both scientifically and application-wise).

P.15 L.350-353: Figure 4 does not show the full picture of the error distribution. It is clearly cut at -14 and 14 TECU. Iif one looks at Figure 3, errors in TEC can easily reach +-20 TECU (just draw a vertical line at any value of Observed TEC, e.g. at 30 TECU). It looks like the authors deliberately try to improve the results of their model performance.

**Minor/Typo comments:**

P.1 L.17: Change "derived" to "obtained"

P.1 L.19: Change "Geomagnetically quiet time (Kp < 3 and Dst > -20 nT) data during the years" to "Data during geomagnetically quiet time (Kp < 3 and Dst > -20 nT) for the years"

P.1 L.22 Change "to obtain the model" to "to obtain model coefficients"

P.1 L.26 Change "COSMIC TEC" to "COSMIC RO TEC"

P.2 L.31: Change "using Global Navigation Satellite Systems" to "in Global Navigation Satellite Systems"

P.2 L.30 Change "during day" to "during the day"

P.2. L.49: Space is missing between "European Geostationary"

P.2 L.50: Change "GPS And Geo-Augmented Navigation" to "GPS-aided Geo Augmented Navigation"

P.3 L.63: Space is missing in "analysis centers"

P.3 L.64: Space is missing in "using the"

P.3 L.64: Change "Global Ionospheric TEC data Map (GIM)" to "Global Ionosphere Maps (GIMs) containing vertical TEC data"

P.3 L.66: Change "Global Ionospheric TEC data Maps (GIMs)" to "GIMs". It has been defined two lines above.

P.3 L.70: Space is missing in "the average"

P.3 L.71: Space is missing in "by CODE"

P.3 L.76: Space is missing in "constructed a"

P.3 L.77: Space is missing in "GPS radio"

P.3 L.82: Space is missing in "related to"

P.4 L.87: Change "localized ionospheric structure" to "localized ionospheric structures"

P.4 L.88: Change "on a global scale model" to "in global models"

P.5 L.140: Space is missing in "during geomagnetically"

P.6 L.147: Change "solar activity" to "solar activity level"

P.6 L.164: Remove "15" in "reduced 15 to 5"

P.7 L.181 Space is missing in "the F10.7"

P.9 L.223: Change "Global Ionosphere Map (GIM) TEC (GIM-TEC)" to "GIM TEC", as it was defined earlier, remove "Center for Orbit Determination in Europe" – it was defined earlier

P.9 L.225-226: Remove "The daily GIM-TEC values are derived using the GNSS data collected from over 200 tracking stations of IGS and other institutions", as this information was given earlier in the text

P.10 L.238: Space is missing in "in turn"

P.14 L.336: Change ";" to ":"

P.14 L.337: Space is missing in "root mean squared"

P.17 L.373: Change ":" to "." In "0.93"

---

## Short Comment (SC2) · 7 Feb 2020

Responses to interactive comments on**, "Modeling Total Electron Content derived from radio occultation measurements by COSMIC satellites over the African Region"**

**By Mungufeni et al.**

February 7, 2020

We thank the anonymous referee for taking time to evaluate our manuscript. All the comments are addressed as shown below.

**Comment:**

The manuscript presents an empirical model describing ionosphere total electron content over African region. Authors use experimental TEC data obtained using dual frequency GNSS RO receivers onboard of COSMIC satellites to construct the model. They validate the model using same type of data that was used to construct the model but for a different period.

**Response:**

In addition to using the same type of data for validating our model, we shall also use TEC measured by ionosonde stations over South Africa. This is in line with comments made by another anonymous reviewer.

**Comment:**

General impression is that the present work has no contribution to the current understanding of the low latitude ionospheric physics/modelling. The work brings a little science and the newly created model could hardly be used in any real-life application. Authors are making too many assumptions and mistakes, sometimes trying to deliberately present performance results better than they are. Moreover, the performance of the model has not been compared to any other well-known model, leaving a room for doubts. Therefore, I recommend the manuscript (in its present form) is **rejected**. At the same time, the work might be improved and worth publication after substantial modifications. Please find below a list of critical issues along with

possible improvements/corrections for a potential future re-submission.

**Response:**

Later comments reveal that the reviewer has kindly elaborated with examples the above comments. Therefore, appropriate responses have been given later following the elaborated comment. Since all comments have been addressed appropriately, we do not expect a decision to reject our manuscript.

**Comment:**

P.1 L.27: Replace "good" with "applied". Otherwise, provide a proof of the model "goodness"

**Response**:

The suggestion will be implemented.

**Comment**:

P.2. L.35-38: Not all GNSS systems support ionospheric corrections. E.g. GLONASS does not broadcast any ionospheric model parameters. Correct the sentence accordingly.

**Response**:

The correction will be done as suggested.

**Comment**:

P.2 L.40: Provide a reference to the original description of Klobuchar model: *"Klobuchar JA (1987) Ionospheric time-delay algorithm for single frequency GPS users. IEEE Trans Aerosp Electron Syst 23(3):325–331. https://doi.org/10.1109/TAES.1987.310829"*

**Response**:

The reference will be added as suggested.

**Comment**:

P.2 L.41-42: NeQuick G model is based on the NeQuick model, but not NeQuick 2. Correct the statement and the reference accordingly, e.g. *"EC (2016) European GNSS (Galileo) Open Service—Ionospheric correction algorithm for Galileo single frequency users, Issue 1.2, Sept. 2016, European Commission"*

**Response**:

The correction will be done and the suggested reference will also be added.

**Comment**:

P.2. L.42: Change "The NeQuick is" to "The NeQuick and its subsequent modifications (NeQuick G and NeQuick 2) are"

**Response**:

The suggestion will be implemented.

**Comment**:

P.2 L.53: IRI model does not provide information about "electron and ion velocities". It only provides information about equatorial vertical ion drift. Correct the sentence accordingly.

**Response**:

The sentence will be corrected as, "For the international standard specification of ionospheric parameters (such as electron density, electron and ion temperatures, and ion drift velocity) …."

**Comment**:

P.2 L.55-56: Change "The model is primarily" to "IRI is an empirical model primarily"

**Response**:

The suggestion will be implemented.

**Comment**:

P.3 L.74: Change "GIM" to "global ionosphere model", as GIM is already defined to be Global Ionosphere  Map.

**Response**:

The suggestion will be implemented.

**Comment**:

P.3 L.76: Change "GIM model" to "global ionosphere model"

**Response**:

The suggestion will be implemented.

**Comment**:

P.3 L.80-82: The high values of RMS in low latitude region provided by CODE is, primarily, due to the inability of the selected model function (spherical harmonics) to describe ionospheric structure in low latitude. Modify the sentence accordingly.

**Response**:

We based on Fig below (obtained from Najman, P. and Kos, T.: Performance Analysis of Empirical Ionosphere Models by Comparison with CODE Vertical TEC Maps, Chapter 13, in: Mitigation of Ionospheric    Threats to GNSS: an Appraisal of the Scientific and Technological Outputs of the TRANSMIT Project, InTech Open Science publications, pp. 162 - 178, doi:10.5772/58774, 2014) to make the statement, "This could be due to the poor distribution of IGS tracking stations over Africa and anomalies in the ionosphere related to the geographic and geomagnetic location".

[Figure]

Mean values of CODE RMS maps of years 2010,2011 and 2012

Indeed, figure above shows high RMS values over the oceans and land masses that have few/no ground based GPS receivers. This situation typically exists around and over the African continent.

Since figure above does not strictly show high values of RMS over all low latitude regions where EIA exists, we intend to remove EIA as a reason for the high RMS values over Africa.

Although the reviewer did not give reference, his/her suggestion of the inability of spherical harmonics to describe well low latitude ionospheric structure is consistent with existence of EIA over low latitudes.

**Comment**:
P.3 L.84: Change "the GIM model" to "global models"

**Response**:

The suggestion will be implemented

**Comment**:
P.4 L.115: Author use TEC integrated up to COSMIC satellite heights (800 km) to construct the model (*"integration being done up to the altitudes of the COSMIC*

*satellites"*). However, the topside TEC values (according to numerous studies, e.g. by Bilitza 2009, Yizengaw 2008 etc.) can reach from 10% to 80% of the total electron content (from ground to GNSS satellite heights). This fact significantly reduces the scientific value and application of the developed model. Essentially, the model is useless for GNSS applications.

**Response**:

We are aware about the existence of substantial ionosphere above the altitude of COSMIC satellites. Concerning the region under study, the upper quartile of the differences between coincident COSMIC RO TEC and ground based GPS TEC could reach ~11 TECU (Mungufeni et al, (2019), *Characterization of Total Electron Content over African region using Radio Occultation observations of COSMIC satellites*, Adv in Space Res 65, 19 – 29). The problem is that such differences vary with location. To make it worse, over the oceans and some land masses such differences may not be established due to lack of ground based GPS TEC over such locations. In general, at the moment it is yet difficult to adjust COSMIC RO TEC to include plasmaspheric TEC. Due to these challenges, we do not trust usage of data from previous studies (e.g 1. Mungufeni et al. (2019), *Estimation of equivalent ground-based total electron content using CHAMP-based GPS observations*, Adv in Space Res 64, 199 – 210 and 2. Okoh, et al. (2019). A neural network based ionospheric model over Africa from COSMIC and Ground GPS observations. Journal of Geophysical Research: Space Physics, 124. https://doi.org/10.1029/2019JA027065) that attempted to scale space based GPS TEC to yield equivalent that would be obtained using ground based GPS TEC.

We thick that lack of inclusion of plasmaspheric TEC in COSMIC RO TEC, does not render the data completely useless. This point can be illustrated by examining the available differences between coincident COSMIC RO TEC and ground based GPS TEC. Since the upper quartiles of the differences can reach up to ~11 TECU, the median/mean values might obviously be much lower than this value. This might be the reason for observing most of the well known ionospheric TEC features over the African region when the COSMIC RO TEC were appropriately binned (Mungufeni et al, (2019), *Characterization of Total Electron Content over African region using Radio*

*Occultationobservations of COSMIC satellites*, Adv in Space Res 65, 19 – 29). The ionospheric features being referred to include; (i) occurrence of minimum and maximum TEC during 0:00–08:00 LT and 12:00–16:00 LT respectively, (ii) occurrence of secondary TEC enhancement (maximum) during 16:00–20:00 LT, (iii) lowest TEC values being observed in June solstice and highest TEC values observed in March equinox, (iv) TEC values increase as solar activity changes from low to high, (v) mid latitude TEC values are lower than those of low latitude regions, and (vi) occurrence of equatorial ionisation anomaly.

Therefore, the current model was built with the aim of simulating these known ionospheric features.

**Comment**:

P.5 L.124-126: This statement "*Since the magnitudes of the TEC obtained from COSMIC occultation 124 measurements are close to ground based GNSS TEC*", is not consistent with the previous statement and studies by Mungufeni et al. 2019. Where they show that, depending on the location, the RMS error can vary from 2 to 8 TECU and error distribution plots show values from -24 to 20 TECU. Such large errors cannot be considered "*close to ground-based GNSS TEC*". Authors, at least, are expected to provide information about relative TEC errors (in %, rather than TECU) to claim that errors can be tolerated (if so).

**Response**:

The response to this comment is similar to that of the previous comment. In particular, we shall mention in the manuscript that the average/median error might be much lower than 11 TECU.

**Comment**:

P.6 L.150: The title of the reference Emmert et al. 2010 is incorrect: *Emmert, J. T., Richmond, A. D., and Drob, D. P.: Statistical analysis of the correlation 412 between the equatorial electrojet and the occurrence of the equatorial ionisation 413 anomaly over the East African sector, J. Geophys. Res., 15; A08322; 414 doi:10.1029/2010JA015326, 2010.*

**Response**:

The correction will be made.

**Comment**:

P.6 L.157-167: The selected spatial resolution of 15º in longitude and 5-8º in latitude is too coarse to describe the ionosphere reasonably, especially for the low latitude region, where TEC is changing dramatically from the crest down/up to two peaks of EIA. E.g. GIM maps (the source of the data for most of the empirical models discussed by the authors in the introductions section) use at least 5º by 2.5º resolution (lon and lat). Moreover, 15º in longitude corresponds to 1 hour in LT. Gradients in TEC as a function of LT during sunrise and sunset hours may reach tens of TECU per hour (e.g. Mungufeni et al. 2019, Fig. 2). Therefore, such coarse spatial resolution in longitude will lead to big errors in the model description.

**Response**:

As already stated in the manuscript (page 6, lines 159 – 160) the problem that might arise when a smaller grid resolution is applied is data gaps in some grids. This problem was also illustrated to another anonymous referee. In the illustration, we showed that there are typically ~80 data points observed in a day over the study area. Obviously this number cannot cover all the 24 hours in a particular grid. The situation becomes worse when more grid shells are created. Therefore our choice of grid resolution ensured a balance between observing fairly fine ionospheric structures and created grid shells are not empty. Another method we used to ensure that the created grid shells are not empty is by binning our data according to 3 different ranges of solar flux levels, instead of binning data according to year.

Although our choice of grid resolution appears to be coarse, we were able to observe the previously mentioned known ionospheric features over the African region. This confirms that the justifications we have given for the choice of $15^o$ longitude (page 6, line 163) and the various latitudinal values (page 6, line 158 and lines 164 - 165) are logical. We have noted a mistake on page 6, line 164. The phrase should be, "… the latitudinal grid resolution was reduced $5^o$ for dip latitude range ….."

**Comment**:

P.6 L.170: The whole solar cycle 24 has relatively low solar activity level compared to the two previous ones. Nevertheless, even if we look only at the 24th solar cycle, 2011 and 2016 could hardly be attributed as years of high solar activity level. Please, modify the statement accordingly (e.g. as it is done on P.7 L.182).

**Response**:

The suggestion will be implemented

**Comment**:

P.7 L.189: Please clarify, how 36 solar flux bins were obtained. From the description, it is only 3 solar flux ranges and 12 months, that gives 36 (3x12). But when listing by a variable, only number 3 has to be specified, as it is done, for example with the rest of the variable (hour, lat and lon). Indeed, if we take 60,480 TEC values indicated in L.189, this number can be obtained by multiplying 5x14x3x12x24, but not 5x14x36x12x24.

**Response**:

In each range of solar flux level, there are 12 nodes, corresponding to the months in a year. We shall state this explicitly in the manuscript

**Comment**:

P.8 L.205: According to the definition of cubic spline, it is a spline constructed of piecewise thirdorder polynomials, meaning none of the B splines used in the model were cubic (order 2 and 4). Change the "cubic B spline" into "B spline of different orders" throughout the text and abstract.

**Response**:

The suggestion will be implemented in the manuscript

**Comment**:

P.9 L.218-220: Consider changing this sentence to something like "In order to assess

the ability of the model to describe the data used to construct the model, modelled data were compared to the experimental one. The results of the self-consistency check are presented in Figure 1."

**Response**:

This suggestion will be implemented in the manuscript

**Comment**:

P.9 L.228-229: It is surprising that the authors compare the results of the climatological model (i.e. model where input data were averaged over time, e.g. one month) with GIM map for a single day of that month. Such a comparison is not correct. On top of that, by looking at TEC maps obtained from COSMIC and later by B spline model (columns 2 and 1), one can hardly see any separation between the peaks of the EIA, that can, taking into account averaging in all the bins (e.g. lat and lon) performed by authors, hardly be comprehended.

**Response**:

The GIM-TEC panel will be removed. This will give more space to include comparisons with other models and ionosonde data

We can see clear separation between the crests of EIA before 17:00 LT. However, after this time (sun set) the crests appear to merge. This is expected as the direction of the zonal electric field reverses at around sunset.

**Comment**:

P.9 L.231-232: By looking at the color plots, a reader can hardly assess the performance of the model. It is suggested, in addition to the plots, to present/discuss the results of the mismodelling in terms of a bias and RMS of the error.

**Response**:

The first intention of presenting figures 1 and 2 was to make readers appreciate the ionospheric features that can be revealed by the data used for modeling. Indeed, we

observed and discussed features such as diurnal, seasonal, and solar flux level dependence to mention.

The second intention was to show that the B spline functions can trace very well the trends in the data used for modeling. Surely, observation of two panels from the same row, but different columns reveals that the B spline function traces the trends in measured data very well.

We would like to mention that detailed validation of our model using independent data was presented in figures 3 and 4. We intend to also validate our model using TEC measured by ionosonde stations over South Africa (a suggestion by another anonymous reviewer).

**Comment**:

P.10 L.250-252: From Fig 1 it cannot be clearly understood the secondary maximum if any, especially at -20 lat. Please, if you discuss a feature, try demonstrating it clearly to the reader. A separate figure, or at least, a dashed line at -20 and 4 in Fig 1 is needed to support the statement.

**Response**:

This suggestion will be implemented

**Comment**:

P.11 L.269-270: In row (b), Fig 1, none of the panel show peaks of the EIA. There is no clear separation of the crest and peaks of EIA. Nor in panels b1/b2 neither in b3. Modify the sentence accordingly.

**Response**:

Using red arrows, we have illustrated in figure below the peaks of the EIA. Attention was given to panels in row (b). The question about separation of the peaks was also raised earlier and it was answered.

[Figure]

(a) LSA Mar TEC at 37.5 ¯E; (i) observed     (ii) modeled     TEC (TECU)

(b) MSA     (iii) GIM-TEC

Peaks of EIA

(c) HSA

Dip Latitude (deg)

LT (hours)

240

**Comment**:

P.277-279: The structure of the crest might differ based on various factors (including level of the geomagnetic disturbance). However, when taken as an average, a clear 2 peak structure is present in low latitudes, representing EIA.

**Response**:

Figure below (Bolaji, et al. 2017. Observations of equatorial ionization anomaly over Africa and Middle East during a year of deep minimum, Ann. Geophys., 35, pp. 123 – 132, 2017) taken during a deep solar minimum still shows several crests of EIA. It should be noted that the horizontal axis represents local time. The figure clearly shows in first panel two crests south of the dip equator.

[Figure]

Another figure showing several crests on either side of the dip equator is below (Mungufeni et al, (2019), *Characterization of Total Electron Content over African region using Radio Occultationobservations of COSMIC satellites*, Adv in Space Res 65, 19 – 29).

[Figure]

Details about the data plotted in panel (a) which clearly shows several crests are as follows. The data consists of average COSMIC RO TEC during 13:00 – 14:00 UT.  The

data were for 67 quiet days in March (2008 - 2015). The F10.7 flux on the days were >120 sfu. The spatial binning resolution was $10^o$ in longitude and $2^o$ in latitude. The data gaps mentioned previously can be seen due to the reduced binning resolution.

To verify these observations of several crests on either side of the dip equator, we might need in situ measurements of electron density by polar orbiting satellites flying at altitude range of 120 – 400 km.

**Comment**:

P.12 L.298-299: The science question in this case is not how to model the observed data, but how to explain the data. What is the physical explanation for the absence of the EIA structure (two peaks and the crest) in TEC values calculated from the ground up to COSMIC satellite heights (~800km). And whether this phenomena is not a limitation of the technique applied to calculate TEC. Namely, TEC computed by integrating electron density profile, that by itself is a product of RO inversion, is subject to big errors, especially in places where big horizontal gradients exist (read, e.g. M.M Shaikh et al., Implementation of Ionospheric Asymmetry Index in TRANSMIT Prototype, DOI: 10.5772/58551). Without understanding the reasons of the observed behavior all the modeling efforts are meaningless.

**Response**:

In the paragraph under question, we mentioned asymmetry of EIA feature and occurrence of secondary peak in TEC over Africa. We further mention that these features can be seen in the data we used to develop our model. Therefore, our model emulates these features. We would like to mention that these two features have been well explained in the manuscript (see page 11, lines 256 – 27 and page 12, lines 284 - 286).

The reviewer's phrase, "absence of the EIA structure (two peaks and the crest) in TEC values calculated from the ground up to COSMIC satellite heights (~800km)" does not

exist in our manuscript. This makes it difficult for us understand the point the reviewer would like to make. Anyway, we guess that the reviewer is talking about absence of asymmetry of EIA feature in GIM-TEC. In case this is correct, the first reason might be poor distribution of ground based GPS receivers over the African region. The second reason as previously stated by the reviewer might be inability of the spherical harmonic function to map TEC over the low latitude regions. We already provided the first reason on page 11, line 253, we shall add the second reason to the manuscript.

**Comment**:

P.13 L.313: One cannot see the "perfect match" of the observed and modelled data just by looking at the plots. At least a third row in form of difference map (error map) has to be presented to visually     assess the error level. Moreover, statistical results (e.g. RMS and bias of the error) must be presented in      order to make such a bold conclusion.

**Response**:

We kindly request the reviewer to have another look at figures 1 and 2, taking for instance two panels from the same row, but different columns. After appreciating the perfect match between the observed and the modeled data, there would be no need for error map.

On the other hand, we understand the importance of error maps, particularly when validating a model with independent set of data. We demonstrated this by presenting figure 4.

**Comment:**

P.13 L.312-324: Authors do not discuss at all the TEC behavior observed in September at lat ~ -20, where its diurnal variation has a maximum during local night hours (21-03 LT). This maximum seems to exceed any other TEC values on this plot (row c, column 1 and 2) and looks like an error in the data processing.      Such behavior seems to have no physical explanation.

**Response**:

On page 13, line 313, we stated that, "among the many features of TEC exhibited …." This means we were interested in the key features. Now that the reviewer has identified a possible out liar during September at lat ~ -20$^o$, we agree to mention the same in the manuscript.

**Comment:**

P.14 Section 5: The authors fail to explain why they need yet another TEC model. Unless the performance of the newly created model is compared to existing models and it is demonstrated that its any better than the rest of the models present on the "ionosphere model market" (e.g. IRI, NeQuick,  NTCM etc.), there is very little value in the study (both scientifically and application-wise).

**Response**:

We shall compare our model with the existing models such as IRI and NeQuick, and AfriTEC (Okoh et al. 2019).

**Comment:**

P.15 L.350-353: Figure 4 does not show the full picture of the error distribution. It is clearly cut at -14 and 14 TECU. Iif one looks at Figure 3, errors in TEC can easily reach +-20 TECU (just draw a vertical line at any value of Observed TEC, e.g. at 30 TECU). It looks like the authors deliberately try to improve the results of their model performance.

**Response:**

Actually, we should have not indicated ±16 on the horizontal axis. Moreover, we should have indicated on the horizontal axis < -14 (instead of merely -14) and > 14 (instead of merely 14). The total number of errors with values in the range of -14 – 14 TECU was 16858 (97.4 %), while the number of errors with values outside this range was 454 (2.6 %). By comparing these two percentages, it can be deduced that the number of errors with values outside the range of -14 – 14 TECU was insignificant. In statistics, conclusions are made based on majority, but not minority.

After implementing the above changes, Figure 4 would appear as below

[Figure]

**Minor/Typo comments:**

P.1 L.17: Change "derived" to "obtained"

P.1 L.19: Change "Geomagnetically quiet time (Kp < 3 and Dst > -20 nT) data during the years" to "Data  during geomagnetically quiet time (Kp < 3 and Dst > -20 nT) for the years"

P.1 L.22 Change "to obtain the model" to "to obtain model coefficients"

P.1 L.26 Change "COSMIC TEC" to "COSMIC RO TEC"

P.2 L.31: Change "using Global Navigation Satellite Systems" to "in Global Navigation Satellite Systems"

P.2 L.30 Change "during day" to "during the day"

P.2. L.49: Space is missing between "European Geostationary"

P.2 L.50: Change "GPS And Geo-Augmented Navigation" to "GPS-aided Geo Augmented Navigation"

P.3 L.63: Space is missing in "analysis centers"

P.3 L.64: Space is missing in "using the"

P.3 L.64: Change "Global Ionospheric TEC data Map (GIM)" to "Global Ionosphere Maps (GIMs)

containing vertical TEC data"

P.3 L.66: Change "Global Ionospheric TEC data Maps (GIMs)" to "GIMs". It has been defined two linesabove.

P.3 L.70: Space is missing in "the average"

P.3 L.71: Space is missing in "by CODE"

P.3 L.76: Space is missing in "constructed a"

P.3 L.77: Space is missing in "GPS radio"

P.3 L.82: Space is missing in "related to"

P.4 L.87: Change "localized ionospheric structure" to "localized ionospheric structures"

P.4 L.88: Change "on a global scale model" to "in global models"

P.5 L.140: Space is missing in "during geomagnetically"

P.6 L.147: Change "solar activity" to "solar activity level"

P.6 L.164: Remove "15" in "reduced 15 to 5"

P.7 L.181 Space is missing in "the F10.7"

P.9 L.223: Change "Global Ionosphere Map (GIM) TEC (GIM-TEC)" to "GIM TEC", as it was defined  earlier, remove "Center for Orbit Determination in Europe" – it was defined earlier

P.9 L.225-226: Remove "The daily GIM-TEC values are derived using the GNSS data collected  from over 200 tracking stations of IGS and other institutions", as this information was given earlier in the text

P.10 L.238: Space is missing in "in turn"

P.14 L.336: Change ";" to ":"

P.14 L.337: Space is missing in "root mean squared"

P.17 L.373: Change ":" to "." In "0.93"

**Response:**

All minor comments will be addressed as suggested by the reviewer

---

## Author Comment (AC1) · 5 Mar 2020

Responses to interactive comments on, **"Modeling Total Electron Content derived from radio occultation measurements by COSMIC satellites over the African Region"**

**By Mungufeni et al.**

March 03, 2020

We thank the reviewers for taking time to evaluate our manuscript. All the comments are addressed as shown below.

**Reviewer 1:**

**Comment:**

An author of this paper (Patrick Mungufeni) along with a long list of other authors have recently published the following paper: Okoh, et al. (2019). A neural network based ionospheric model over Africa from COSMIC and Ground GPS observations. Journal of Geophysical Research: Space Physics, 124. https://doi.org/10.1029/2019JA027065. In that particular paper the authors perform an adjustment using Neural Networks according to which they correct the reasonable discrepancy between TEC from ground based receivers (up to 22000 Km) and occultation measurements up to 700 Km. They seem to apply no such procedure in this paper. This is a major problem of this paper. They also need to make special reference to that paper.

**Response:**

Indeed, Patrick Mungufeni contributed to the paper in the comment which was published on Thur, Dec 12, 5:23 PM. The current manuscript under discussion was submitted on Saturday, Nov 23, 2:09 PM (Korean time). Therefore, we could not reference Okoh et al, (2019) since it was published later after the current submission. Below are the screen shots of emails to prove the dates. Anyway, we shall reference Okoh et al. (2019).

[Figure]

Although the reviewer is recommending creation of data base consisting of both ground and space based TEC measurements, such data base may be subjected to criticism. For example, the observation in Okoh et al. (2019) where the ratio between ground

based and COSMIC TEC varies spatially implies that neural network may not learn the relationship between the two data sets over locations which only have COSMIC TEC data. We have highlighted with magenta boxes in Figure below such regions which mostly have COSMIC TEC. The Figure was taken from Okoh et al. (2019). Over the boxes, the adjustments made to COSMIC TEC may not be trusted because of large distances over which interpolations are done.

[Figure]

When a study opts to have both adjusted COSMIC TEC data and ground based GPS TEC data, some locations will be represented by adjusted COSMIC TEC (remember not trusted) while others will be represented by ground based GPS TEC. Obviously, there is still disparity. For purposes of consistency, it might be fair to use entirely adjusted

COSMIC TEC since it can also be available where there is ground based GPS TEC. Since we do not trust the current known procedures for adjusting space based observations (Okoh et al. (2019) and Mungufeni et al. (2019), *Estimation of equivalent ground-based total electron content using CHAMP-based GPS observations*, Adv in Space Res 64, 199 - 210) the current manuscript used only COSMIC TEC without any adjustment.

**Comment:**

Maybe they should compare the output of the NN model out of that paper with the output of the spline model for this paper despite that the COSMIC dataset is used as a basis for both models. In this way they will prove their approach for this paper (omitting any correction for the plasmaspheric contribution which is expected to be high at middle African latitudes

**Response:**

The suggestion in the comment will be implemented. As indicated in Okoh et al. 2019, TEC plots based on NN model can be obtained from MATLAB Central website

(https://www.mathworks.com/matlabcentral/fileexchange/69257-african-gnss-tec-afritec-model?s_tid=prof_contriblnk).   We present in Figure below an example of TEC generated by NN model on March 21, 2012 at 11 UT.

[Figure]

The corresponding TEC map generated based on our model (spline method) is presented below.

[Figure]

In the two TEC maps above, it can be seen that unlike our TEC map which shows the EIA trough (see magenta arrows), the NN TEC map failed to capture the EIA trough. This short fall in NN model TEC map might be due to poor data to represent day of year during model development. This point has been illustrated properly in the response to the next comment. It would be interesting to compare error levels produced when some measured TEC is compared with (i) NN TEC map and (ii) our spline model. We may not perform such analysis since model in (i) is based on electron density that is integrated from ground up to GPS satellites, while model in (ii) is based on electron density integrated up to ~800 km.

**Comment:**

The authors do not provide any scheme by which they would reject any unrealistic COSMIC profiles. There have been numerous validation studies with Digisondes that verify this problem especially in the bottomside.

**Response**:

Empirical modeling requires adequate data for the mathematical functions to capture the physics inherent in the data. However, to minimize measurement errors, studies that have used COSMIC data commonly reject measurements with horizontal smear > 1500 km. We have presented in Figure below the number of COSMIC TEC measurements per day during the year 2013 over the longitude and latitude ranges of -15 – 60$^\circ$ and -35 – 35$^\circ$, respectively.

[Figure]

The blue dots indicate COSMIC TEC measurements when the horizontal smear is < 1500 km, while the red stars indicate COSMIC TEC measurements without limitation of horizontal smear. It can be noticed that when the horizontal smear is limited, ~40 observations may be made per day. Obviously, the 40 measurements may not cover very well all the 24 hours in a day and all the grid cells. This shows clearly that the seasonal or day number of year variation does not have good input data for the entire African region. This is clearly manifested in NN TEC model presented previously. In order to have fairly adequate data, we did not apply restriction to the horizontal smear. Therefore, in a day, there were about 80 observations as shown with red stars in Figure above.

We established that the COSMIC TEC data values with smear > 1500 km do not introduce alarming errors. This was done by analyzing COSMIC TEC data which were coincident with TEC observed by ionosonde stations at Hermanus, Grahamstown, and Louisvale. The observations of the year 2013 were considered. Table below presents

the root mean squared error between (i) ionosonde and COSMIC TEC without limiting the horizontal smear, and (ii) ionosonde and COSMIC TEC with horizontal smear limited to 1500 km.

| Station | Smear < 1500 km | | No limitation | |
|---|---|---|---|---|
| | Number of observations | RMSE (TECU) | Number of observations | RMSE (TECU) |
| Hermanus | 38 | 1.838 | 65 | 2.256 |
| Grahamstown | 34 | 6.479 | 73 | 7.923 |
| Louisvale | 42 | 2.765 | 91 | 3.252 |

The table shows that the RMSE for the two cases over a particular ionosonde station are not grossly different. Based on these results, trading off accuracy may not be costly compared to trading off adequate need of data. Therefore, we decided not to impose any restriction on the horizontal smear. Although the RMSE appear to be smaller when the smear < 1500 km, some of the data points that were subjected to this restriction are also far from the linear least squares fitting line. See blue stars in Figure below.

[Figure]

Most likely, the ~80 COSMIC TEC data points in a day may not still cover very well all the 24 hours in a day and all the grid cells. This problem might be solved by adopting appropriate data binning criteria. Therefore, instead of binning data according to year, we binned data according to only three different solar flux levels. This technique proved to be good and it was published in Mungufeni et al, (2019), *Characterization of Total Electron Content over African region using Radio Occultationobservations of COSMIC satellites*, Adv in Space Res 65, 19 – 29.

**Comment:**

I strongly suggest to compare the output of their model with ionospheric TEC (up to 700 km) from all over four stations Digisonde stations over South Africa https://spaceweather.sansa.org.za/products-andservices/current-conditions/ionograms. This will provide a much more realistic comparison test to their model

**Response:**

The suggestion in the comment will be implemented. We illustrate in Figure below with continuous green lines the diurnal patterns of TEC measured by ionosonde stations at Grahamstown, Hermanus, and Lousville. The corresponding TEC generated by our model and Nequick are superimposed with crosses and diamonds, respectively. We need to mention that during computation of TEC using Nequick, the height was limited to the approximate altitude of the COSMIC satellites (800 km). The panels in columns (i) and (ii) show TEC on DOY 160 (June) and 260 (September), respectively. Both days are of the year 2013. Preliminarily, there appears to be good correspondence between the observed and the modeled TEC. We intend to generate such data plotted in the figures for the entire year 2013 and then perform statistical analysis of the differences between the observed and the model TEC data.

[Figure]

**Reviewer 2**

**Comment:**

The manuscript presents an empirical model describing ionosphere total electron content over African region. Authors use experimental TEC data obtained using dual frequency GNSS RO receivers onboard of COSMIC satellites to construct the model. They validate the model using same type of data that was used to construct the model but for a different period.

**Response:**

In addition to using the same type of data for validating our model, we shall also use TEC measured by ionosonde stations over South Africa. This has been demonstrated in the response to the previous comment.

**Comment:**

General impression is that the present work has no contribution to the current understanding of the low latitude ionospheric physics/modelling. The work brings a little science and the newly created model could    hardly    be    used    in    any    real-life application. Authors are making too many assumptions and mistakes, sometimes trying to deliberately present performance results better than they are. Moreover, the performance of the model has not been compared to any other well-known model, leaving a room for   doubts. Therefore, I recommend the manuscript (in its present form) is **rejected**. At the same time, the  work  might  be  improved  and  worth  publication after substantial modifications. Please find below a list of     critical   issues   along   with possible improvements/corrections for a potential future re-submission.

**Response:**

Later comments reveal that the reviewer has kindly elaborated with examples of the above comments. Therefore, appropriate responses have been given later following the elaborated comments. Since all comments have been addressed appropriately, we do not expect a decision to reject our manuscript.

**Comment:**
P.1 L.27: Replace "good" with "applied". Otherwise, provide a proof of the model "goodness"

**Response**:

The suggestion will be implemented.

**Comment**:
P.2. L.35-38: Not all GNSS systems support ionospheric corrections. E.g. GLONASS

does not broadcast any ionospheric model parameters. Correct the sentence accordingly.

**Response**:

The correction will be done as suggested.

**Comment**:

P.2 L.40: Provide a reference to the original description of Klobuchar model: *"Klobuchar JA (1987) Ionospheric time-delay algorithm for single frequency GPS users. IEEE Trans Aerosp Electron Syst 23(3):325–331. https://doi.org/10.1109/TAES.1987.310829"*

**Response**:

The reference will be added as suggested.

**Comment**:

P.2 L.41-42: NeQuick G model is based on the NeQuick model, but not NeQuick 2. Correct the statement and the reference accordingly, e.g. *"EC (2016) European GNSS (Galileo) Open Service—Ionospheric correction algorithm for Galileo single frequency users, Issue 1.2, Sept. 2016, European Commission"*

**Response**:

The correction will be done and the suggested reference will also be added.

**Comment**:

P.2. L.42: Change "The NeQuick is" to "The NeQuick and its subsequent modifications (NeQuick G and NeQuick 2) are"

**Response**:

The suggestion will be implemented.

**Comment**:

P.2 L.53: IRI model does not provide information about "electron and ion velocities". It

only provides information about equatorial vertical ion drift. Correct the sentence accordingly.

**Response**:

The sentence will be corrected as, "For the international standard specification of ionospheric parameters (such as electron density, electron and ion temperatures, and ion drift velocity) …."

**Comment**:
P.2 L.55-56: Change "The model is primarily" to "IRI is an empirical model primarily"

**Response**:

The suggestion will be implemented.

**Comment**:
P.3 L.74: Change "GIM" to "global ionosphere model", as GIM is already defined to be Global Ionosphere Map.

**Response**:

The suggestion will be implemented.

**Comment**:
P.3 L.76: Change "GIM model" to "global ionosphere model"

**Response**:

The suggestion will be implemented.

**Comment**:
P.3 L.80-82: The high values of RMS in low latitude region provided by CODE is, primarily, due to the inability of the selected model function (spherical harmonics) to describe ionospheric structure in low latitude. Modify the sentence accordingly.

**Response**:

We based on Fig below (obtained from Najman, P. and Kos, T.: Performance Analysis of Empirical Ionosphere Models by Comparison with CODE Vertical TEC Maps, Chapter 13, in: Mitigation of Ionospheric Threats to GNSS: an Appraisal of the Scientific and Technological Outputs of the TRANSMIT Project, InTech Open Science publications, pp. 162 - 178, doi:10.5772/58774, 2014) to make the statement, "This could be due to the poor distribution of IGS tracking stations over Africa and anomalies in the ionosphere related to the geographic and geomagnetic location".

[Figure]

Indeed, figure above shows high RMS values over the oceans and land masses that have few/no ground based GPS receivers. This situation typically exists around and over the African continent.

Since figure above does not strictly show high values of RMS over all low latitude regions where EIA exists, we intend to remove EIA as a reason for the high RMS values over Africa.

Although the reviewer did not give reference, his/her suggestion of the inability of spherical harmonics to describe well low latitude ionospheric structure is consistent with existence of EIA over low latitudes.

**Comment**:

P.3 L.84: Change "the GIM model" to "global models"

**Response**:

The suggestion will be implemented

**Comment**:

P.4 L.115: Author use TEC integrated up to COSMIC satellite heights (800 km) to construct the model (*"integration being done up to the altitudes of the COSMIC satellites"*). However, the topside TEC values (according to numerous studies, e.g. by Bilitza 2009, Yizengaw 2008 etc.) can reach from 10% to 80% of the total electron content (from ground to GNSS satellite heights). This fact significantly reduces the scientific value and application of the developed model. Essentially, the model is useless for GNSS applications.

**Response**:

We think that lack of inclusion of plasmaspheric TEC in COSMIC RO TEC, does not render the data completely useless. This point can be illustrated by examining the available differences between coincident COSMIC RO TEC and ground based GPS TEC. Concerning the region under study, the upper quartile of the differences between coincident COSMIC RO TEC and ground based GPS TEC could reach ~11 TECU (Mungufeni et al, (2019), *Characterization of Total Electron Content over African region using Radio Occultation observations of COSMIC satellites*, Adv in Space Res 65, 19 – 29). Since the upper quartiles of the differences can reach up to ~11 TECU, the median/mean values might obviously be much lower than this value. This might be the reason for observing most of the well known ionospheric TEC features over the African region when the COSMIC RO TEC were appropriately binned in the above reference. The ionospheric features being referred to include; (i) occurrence of minimum and maximum TEC during 0:00–08:00 LT and 12:00–16:00 LT respectively, (ii) occurrence of secondary TEC enhancement (maximum) during 16:00–20:00 LT, (iii) lowest TEC values being observed in June solstice and highest TEC values observed in March equinox, (iv) TEC values increase as solar activity changes from low to high, (v)

mid latitude TEC values are lower than those of low latitude regions, and (vi) occurrence of equatorial ionization anomaly.

Therefore, the current model was built with the aim of simulating these known ionospheric features. Such simulations are important for education purposes. Moreover we illustrated in a previous response that our model generates TEC values consistently with that observed by ionosondes. This implies that equivalent TEC measured by ionosondes over locations which do not have ionosondes can be predicted fairly well using our model.

**Comment**:

P.5 L.124-126: This statement "*Since the magnitudes of the TEC obtained from COSMIC occultation 124 measurements are close to ground based GNSS TEC*", is not consistent with the previous statement and studies by Mungufeni et al. 2019. Where they show that, depending on the location, the RMS error can vary from 2 to 8 TECU and error distribution plots show values from -24 to 20 TECU. Such large errors cannot be considered "*close to ground-based GNSS TEC*". Authors, at least, are expected to provide information about relative TEC errors (in %, rather than TECU) to claim that errors can be tolerated (if so).

**Response**:

This comment is similar to the previous one that was already responded to. Much as it appears good to report percentages of the error levels, it needs to be noted that only relying on percentages may be misleading. For example, 90 % which looks impressive can be obtained from 9 samples drawn from 10 observations. Statistically, the number of observations is too small to be relied on in order to draw a general conclusion. In addition to the number of different error levels reported in the stated references, we shall go back to analyze the data presented in the stated reference and also report the percentages of the different error levels.

**Comment**:

P.6 L.150: The title of the reference Emmert et al. 2010 is incorrect:
*Emmert, J. T., Richmond, A. D., and Drob, D. P.: Statistical analysis of the correlation*

*412 between the equatorial electrojet and the occurrence of the equatorial ionisation*
*413 anomaly over the East African sector, J. Geophys. Res., 15; A08322; 414*
*doi:10.1029/2010JA015326, 2010.*

**Response**:

The correction will be made.

**Comment**:

P.6 L.157-167: The selected spatial resolution of 15º in longitude and 5-8º in latitude is too coarse to describe the ionosphere reasonably, especially for the low latitude region, where TEC is changing dramatically from the crest down/up to two peaks of EIA. E.g. GIM maps (the source of the data for most of the empirical models discussed by the authors in the introductions section) use at least 5º by 2.5º resolution (lon and lat). Moreover, 15º in longitude corresponds to 1 hour in LT. Gradients in TEC as a function of LT during sunrise and sunset hours may reach tens of TECU per hour (e.g. Mungufeni et al. 2019, Fig. 2). Therefore, such coarse spatial resolution in longitude will lead to big errors in the model description.

**Response**:

We have tried to develop the model using data binned in grids with longitude range of 5º and latitude range of 3º. Indeed, there is improvement in the output as shown in the figures presented in this document. With the exception of the last figure in this document, all the model outputs presented in this document were based on the new model.

**Comment**:

P.6 L.170: The whole solar cycle 24 has relatively low solar activity level compared to the two previous ones. Nevertheless, even if we look only at the 24th solar cycle, 2011 and 2016 could hardly be attributed as years of high solar activity level. Please, modify the statement accordingly (e.g. as it is done on P.7 L.182).

**Response**:

The sentence will be modified as, "Since there were many geomagnetically disturbed days during the years (2011 - 2016), ……".

**Comment**:

P.7 L.189: Please clarify, how 36 solar flux bins were obtained. From the description, it is only 3 solar flux ranges and 12 months, that gives 36 (3x12). But when listing by a variable, only number 3 has to be specified, as it is done, for example with the rest of the variable (hour, lat and lon). Indeed, if we take 60,480 TEC values indicated in L.189, this number can be obtained by multiplying 5x14x3x12x24, but not 5x14x36x12x24.

**Response**:

In each of the 3 solar flux ranges, there are 12 nodes, corresponding to the months in a year (see table 2 and page 7, line 184). This results in to the 36 solar flux bins. The listing in equation 2 which appears as 3 will be changed to 36.

**Comment**:

P.8 L.205: According to the definition of cubic spline, it is a spline constructed of piecewise thirdorder polynomials, meaning none of the B splines used in the model were cubic (order 2 and 4). Change the "cubic B spline" into "B spline of different orders" throughout the text and abstract.

**Response**:

The suggestion will be implemented.

**Comment**:

P.9 L.218-220: Consider changing this sentence to something like "In order to assess the ability of the model to describe the data used to construct the model, modelled data were compared to the experimental one. The results of the self-consistency check are presented in Figure 1."

**Response**:

This suggestion will be implemented in the manuscript

**Comment**:

P.9 L.228-229: It is surprising that the authors compare the results of the climatological model (i.e. model where input data were averaged over time, e.g. one month) with GIM map for a single day of that month. Such a comparison is not correct. On top of that, by looking at TEC maps obtained from COSMIC and later by B spline model (columns 2 and 1), one can hardly see any separation between the peaks of the EIA, that can, taking into account averaging in all the bins (e.g. lat and lon) performed by authors, hardly be comprehended.

**Response**:

The GIM-TEC panel will be removed.

After reducing the spatial resolution of the data, figure below (modified version of the figure being referred to in the comment) shows distinct two crests and a trough.

**Comment**:

P.9 L.231-232: By looking at the color plots, a reader can hardly assess the performance of the model. It is suggested, in addition to the plots, to present/discuss the results of the mismodelling in terms of a bias and RMS of the error.

**Response**:

The suggestion in the comment will be implemented. See third column panels in Figure below. The error map shows error levels mostly < 0.5 TECU. We shall also perform statistical analysis of the error levels as recommended by the reviewer.

**Comment**:

P.10 L.250-252: From Fig 1 it cannot be clearly understood the secondary maximum if any, especially at -20 lat. Please, if you discuss a feature, try demonstrating it clearly to the reader. A separate figure, or at least, a dashed line at -20 and 4 in Fig 1 is needed to support the statement.

**Response**:

This suggestion will be implemented as demonstrated in Figure below. The two arrows in magenta color exist a long a particular latitude. The two different peaks in diurnal TEC along the arrows can be seen.

[Figure]

**Comment**:

P.11 L.269-270: In row (b), Fig 1, none of the panel show peaks of the EIA. There is no clear separation of the crest and peaks of EIA. Nor in panels b1/b2 neither in b3. Modify the sentence accordingly.

**Response**:

The above figure clearly shows the two different EIA peaks. Particularly, after 14:00 LT when EIA formation is at its maximum.

**Comment**:

P.277-279: The structure of the crest might differ based on various factors (including level of the geomagnetic disturbance). However, when taken as an average, a clear 2 peak structure is present in low latitudes, representing EIA.

**Response**:

The remark in the comment will be used to adjust the discussions on page 7, lines 275 - 281.

**Comment**:

P.12 L.298-299: The science question in this case is not how to model the observed data, but how to explain the data. What is the physical explanation for the absence of the EIA structure (two peaks and the crest) in TEC values calculated from the ground up to COSMIC satellite heights (~800km).   And whether this phenomena is not a limitation of the technique applied to calculate TEC. Namely,   TEC computed by integrating electron density profile, that by itself is a product of RO inversion, is subject to big errors, especially in places where big horizontal gradients exist (read, e.g. M.M Shaikh  et al., Implementation of Ionospheric Asymmetry Index in TRANSMIT Prototype, DOI:   10.5772/58551). Without understanding the reasons of the observed behavior all the modeling efforts are meaningless.

**Response**:

In the paragraph under question, we mentioned asymmetry of EIA feature and occurrence of secondary peak in TEC over Africa. We further mention that these features can be seen in the data we used to develop our model. Therefore, our model emulates these features. We would like to mention that these two features have been well explained in the manuscript (see page 11, lines 256 – 27 and page 12, lines 284 - 286).

The reviewer's phrase, "absence of the EIA structure (two peaks and the crest) in TEC values calculated from the ground up to COSMIC satellite heights (~800km)" does not exist in our manuscript. This makes it difficult for us to understand the point the reviewer would like to make. Anyway, we guess that the reviewer is talking about the absence of asymmetry of EIA feature in GIM-TEC. In case this is correct, the first reason might be poor distribution of ground based GPS receivers over the African region. The second reason as previously stated by the reviewer might be inability of the spherical harmonic function to map TEC over the low latitude regions. We already provided the first reason on page 11, line 253, we shall add the second reason to the manuscript.

**Comment**:

P.13 L.313: One cannot see the "perfect match" of the observed and modelled data just by looking at the plots. At least a third row in form of difference map (error map) has to be presented to visually    assess the error level. Moreover, statistical results (e.g. RMS and bias of the error) must be presented in    order to make such a bold conclusion.

**Response**:

We have provided in the third column panels of previous Figure the error levels between the observed and the measured data. The error maps mostly show values less than 0.5 TECU (see the color bar). This means as expected, the observed and the modeled data almost match perfectly. The Figure mentioned in the comment will be modified as suggested by the reviewer. Moreover, we shall also present the statistics of the error levels.

**Comment:**

P.13 L.312-324: Authors do not discuss at all the TEC behavior observed in September at lat ~ -20, where its diurnal variation has a maximum during local night hours (21-03 LT). This maximum seems to exceed any other TEC values on this plot (row c, column 1 and 2) and looks like an error in the data processing.    Such behavior seems to have no physical explanation.

**Response**:

On page 13, line 313, we stated that, "among the many features of TEC exhibited …." This means we were interested in the key features. Anyway, after carefully analyzing the data again the feature mentioned in the comment was no longer seen. We intend to provide details of the analysis in the revised manuscript.

**Comment:**

P.14 Section 5: The authors fail to explain why they need yet another TEC model. Unless the performance of the newly created model is compared to existing models and it is demonstrated that its any better than the rest of the models present on the "ionosphere model market" (e.g. IRI, NeQuick, NTCM etc.), there is very little value in the study (both scientifically and application-wise).

**Response**:

We shall compare our model with the existing models such as IRI, NeQuick, and AfriTEC (Okoh et al. 2019). Examples of the comparisons were already shown previously in this same document. The examples where we compared TEC output of our model with that measured by ionosondes seem to indicate that our model predicts well. Being able to predict well TEC measured by ionosondes is important in predicting such quantities over locations where there are no ionosondes. This might be the scientific significance of this study. Another scientific significance of this study is the demonstration of the potential of spline functions to model TEC over the African region.

**Comment:**

P.15 L.350-353: Figure 4 does not show the full picture of the error distribution. It is clearly cut at -14 and 14 TECU. Iif one looks at Figure 3, errors in TEC can easily reach +-20 TECU (just draw a vertical line at any value of Observed TEC, e.g. at 30 TECU). It looks like the authors deliberately try to improve the results of their model performance.

**Response:**

Actually, we should have not indicated ±16 on the horizontal axis. Moreover, we should have indicated on the horizontal axis < -14 (instead of merely -14) and > 14 (instead of

merely 14). The total number of errors with values in the range of -14 – 14 TECU was 16858 (97.4 %), while the number of errors with values outside this range was 454 (2.6 %). By comparing these two percentages, it can be deduced that the number of errors with values outside the range of -14 – 14 TECU was insignificant. After implementing the above changes, Figure 4 would appear as below. A long side this figure, we shall also present the percentages of the different error levels. This is in accordance with the previous comment of the reviewer.

[Figure]

**Minor/Typo comments:**

P.1 L.17: Change "derived" to "obtained"

P.1 L.19: Change "Geomagnetically quiet time (Kp < 3 and Dst > -20 nT) data during the years" to "Data during geomagnetically quiet time (Kp < 3 and Dst > -20 nT) for the years"

P.1 L.22 Change "to obtain the model" to "to obtain model coefficients"

P.1 L.26 Change "COSMIC TEC" to "COSMIC RO TEC"

P.2 L.31: Change "using Global Navigation Satellite Systems" to "in Global Navigation Satellite Systems"

P.2 L.30 Change "during day" to "during the day"

P.2. L.49: Space is missing between "European Geostationary"

P.2 L.50: Change "GPS And Geo-Augmented Navigation" to "GPS-aided Geo Augmented Navigation"

P.3 L.63: Space is missing in "analysis centers"

P.3 L.64: Space is missing in "using the"

P.3 L.64: Change "Global Ionospheric TEC data Map (GIM)" to "Global Ionosphere Maps (GIMs)

containing vertical TEC data"

P.3 L.66: Change "Global Ionospheric TEC data Maps (GIMs)" to "GIMs". It has been defined two linesabove.

P.3 L.70: Space is missing in "the average"

P.3 L.71: Space is missing in "by CODE"

P.3 L.76: Space is missing in "constructed a"

P.3 L.77: Space is missing in "GPS radio"

P.3 L.82: Space is missing in "related to"

P.4 L.87: Change "localized ionospheric structure" to "localized ionospheric structures"

P.4 L.88: Change "on a global scale model" to "in global models"

P.5 L.140: Space is missing in "during geomagnetically"

P.6 L.147: Change "solar activity" to "solar activity level"

P.6 L.164: Remove "15" in "reduced 15 to 5"

P.7 L.181 Space is missing in "the F10.7"

P.9 L.223: Change "Global Ionosphere Map (GIM) TEC (GIM-TEC)" to "GIM TEC", as it was defined  earlier, remove "Center for Orbit Determination in Europe" – it was defined earlier

P.9 L.225-226: Remove "The daily GIM-TEC values are derived using the GNSS data collected  from over 200 tracking stations of IGS and other institutions", as this information was given earlier in the text

P.10 L.238: Space is missing in "in turn"

P.14 L.336: Change ";" to ":"

P.14 L.337: Space is missing in "root mean squared"

P.17 L.373: Change ":" to "." In "0.93"

**Response:**

All minor comments will be addressed as suggested by the reviewer

---

## Author Response (AR1)

Responses to referee comments on, **"Modeling Total Electron Content derived from radio occultation measurements by COSMIC satellites over the African Region"**

**By Mungufeni et al.**

May 03, 2020

We thank the editor and reviewers for taking time to evaluate our manuscript. All the comments are addressed as shown below.

**Editor:**

Thank you for submitting your manuscript to Annales Geophysicae and for your response to the referees' comments. The study you are reporting is interesting contribution to the knowledge of the coupling of the solar wind-magnetosphere-ionosphere system over the African region. However, as you already know, the referees had some important objections to the present version of the manuscript, and this is a reason why I am suggesting a major revision. From my side, I suggest to make reference in the improved manuscript the already published paper by Okoh et al (2019) and introduce brief comparison of the approaches and finding presented in both your and Okoh's paper. Please, consider carefully and discuss in the revised version of the manuscript all comments of the referees indicating the changes you have newly introduced. The manuscript will be revised once again. If you are prepared to undertake the improvements required, please submit the revised manuscript.

**Response:**

*The paper by Okoh et al., (2019) has been cited and discussed in the revised version of the manuscript on page 4, lines 15 – 23; page 5, lines 12 – 22; page 8, lines 2 – 7; pages 22 – 24, page 29, lines 14 - 18. Comparison of approaches and findings presented in our paper and that of Okoh et al., (2019) can be deduced from text on page 4, lines 15 – 26; page 5, lines 12 – 22; and pages 22 - 24. The highlights of the comparisons and findings are as follows.*

| Differences between approaches in Okoh et al., (2019) and the current study | | |
|---|---|---|
| | Okoh et al., (2019) | Current study |
| 1 | used neural network technique to develop TEC model over the entire African region. | used B spline functions to model TEC over the entire African region. |
| 2 | Used both adjusted COSMIC RO TEC and ground-based GPS TEC | Used only COSMIC RO TEC without adjustment |

Findings:

Due to the lack of a dense network of ground-based GNSS receivers and poor coverage of COSMIC RO data over the African region, the TEC model over the entire African region presented by Okoh et al. (2019) sometimes failed to capture the equatorial ionization anomaly (EIA) over the region. This point has been illustrated with examples in this document on page 7. To overcome poor coverage of COSMIC RO data, the current study applied data binning method that allowed development of an improved TEC model over the region. Another reason we suspect that might have resulted in TEC maps based on Okoh et al., (2019) not yielding the EIA feature sometimes could be discrepancy between two data sets which were used. i.e adjusted COSMIC RO TEC and ground-based GPS TEC. We illustrate in this document on pages 5 and 6 the possible reason for the shortfall in the adjustment of COSMIC RO TEC. Based on these issues, this study only used COSMIC RO TEC without adjustment. Since this data comprised of electron density up to ~800 km, our model was able to reproduce equivalent TEC observed by ionosonde stations. On the other hand, the model of Okoh et al., (2019) can yield equivalent TEC observed by ground-based GPS receivers.

Comments by the reviewers are addressed as shown below.

**Responses to Reviewer #1**

**Comment:**

An author of this paper (Patrick Mungufeni) along with a long list of other authors have recently published the following paper: Okoh, et al. (2019). A neural network based ionospheric model over Africa from COSMIC and Ground GPS observations. Journal of Geophysical Research: Space Physics, 124. https://doi.org/10.1029/2019JA027065. In that particular paper the authors perform an adjustment using Neural Networks according to which they correct the reasonable discrepancy between TEC from ground based receivers (up to 22000 Km) and occultation measurements up to 700 Km. They seem to apply no such procedure in this paper. This is a major problem of this paper. They also need to make special reference to that paper.

**Response:**

*The preceding response (to the editor) specifies page and line numbers where citations and discussions of Okoh et al. (2019) within the current version of the manuscript can be seen. The numerous cases of citations and discussions signify a special mention of Okoh et al. (2019). While we agree to the reviewer's recommendation to create a data base consisting of both ground and space based TEC measurements, such data base may be subjected to several other issues and criticisms. These issues have been discussed in the revised manuscript in page 5, lines 12 – 22; page 8, lines 5 – 7.*

*More explanations which could not be put in the manuscript due to space constraints are provided below.*

Indeed, Patrick Mungufeni contributed to the paper in the comment which was published on Thur, Dec 12, 5:23 PM. The current manuscript was first submitted on Saturday, Nov 23, 2:09 PM (Korean time). Therefore, we could not reference Okoh et al,

(2019) since it was published later after our first submission. Below are the screen shots of emails to prove the dates.

[Figure]

[Figure]

Although the reviewer is recommending creation of data base consisting of both ground and space based TEC measurements, such data base may be subjected to criticism. For example, the observation in Okoh et al. (2019) where the ratio between ground based and COSMIC TEC varies spatially implies that neural network may not learn the relationship between the two data sets over locations which only have COSMIC TEC data. We have highlighted with magenta boxes in Figure below such regions which mostly have COSMIC TEC. The Figure was taken from Okoh et al. (2019). Over the boxes, the adjustments made to COSMIC TEC may not be trusted because of large distances over which interpolations are done.

[Figure]

When a study opts to have both adjusted COSMIC TEC data and ground based GPS TEC data, some locations will be represented by adjusted COSMIC TEC (remember not trusted) while others will be represented by ground based GPS TEC. Obviously, there is still disparity. For purposes of consistency, it might be fair to use entirely adjusted COSMIC TEC since it can also be available where there is ground based GPS TEC. Unlike in Okoh et al. (2019) and Mungufeni et al. (2019), the current manuscript used only COSMIC TEC without any adjustment.

**Comment:**

May be they should compare the output of the NN model out of that paper with the output of the spline model for this paper despite that the COSMIC dataset is used as a basis for both models. In this way they will prove their approach for this paper (omitting any correction for the plasmaspheric contribution which is expected to be high at middle African latitudes.

**Response:**

*In pages 22 - 24, we have presented comparison between NN TEC maps and spline technique TEC maps.* As indicated in Okoh et al. 2019, TEC plots based on NN model can be obtained from MATLAB Central website (https://www.mathworks.com/matlabcentral/fileexchange/69257-african-gnss-tec-afritec-model?s_tid=prof_contriblnk). We present in Figure below examples of TEC generated by NN model at 11 UT on DOY 81 (March), 171 (June), 260 (September), and 347 (December) of the year 2012.

[Figure]

The corresponding TEC maps generated based on our model (spline method) are presented below.

[Figure]

Unlike our TEC maps which clearly show the EIA trough (see magenta arrows) in all the seasons, the neural network technique TEC maps only clearly capture the EIA trough in December solstice. As pointed before, this short fall in neural network TEC model might be due to poor amount of data to represent day of year during model development and discrepancy between the two data sets (adjusted COSMIC RO TEC and ground-based GPS TEC) used in the study. Another observation that can be made from above two sets of figures is that unlike the neural network model which yields smooth spatial TEC variation, the spline modeling technique does not yield smooth spatial TEC variation. In real life, measurement or observed values rarely vary smoothly. Since the spline modeling technique produces results (see Figure 1 of revised manuscript) which

demonstrate that the modeled data matches almost perfectly the observed data, it is expected that the spatial variations of TEC in maps of spline model are not smooth.

It would be interesting to compare error levels produced when some measured TEC is compared with (i) NN TEC map and (ii) our spline model. We may not perform such analysis since model in (i) is based on electron density that is integrated from ground up to GPS satellites, while model in (ii) is based on electron density integrated up to ~800 km.

**Comment:**

The authors do not provide any scheme by which they would reject any unrealistic COSMIC profiles. There have been numerous validation studies with Digisondes that verify this problem especially in the bottomside.

**Response**:

*Justifications for not rejecting some of the values are discussed in page 7, lines 14 – 18, page 8, lines 1 - 22. Due to space limits in the manuscript, we could not include the following analysis.*

Empirical modeling requires adequate data for the mathematical functions to capture the physics inherent in the data. However, to minimize measurement errors, studies that have used COSMIC data commonly reject measurements with horizontal smear > 1500 km. We have presented in Figure below the number of COSMIC TEC measurements per day during the year 2013 over the longitude and latitude ranges of -15 – 60$^o$ and -35 – 35$^o$, respectively.

[Figure]

The blue dots indicate COSMIC TEC measurements when the horizontal smear is < 1500 km, while the red stars indicate COSMIC TEC measurements without limitation of horizontal smear. It can be noticed that when the horizontal smear is limited, ~40 observations may be made per day. Obviously, the 40 measurements may not cover very well all the 24 hours in a day and all the grid cells. In the revised manuscript, there are 9,216 (16 longitudinal, 24 latitudinal, and 24 local time) grid cells arising from 5° lon, 3° lat, and 1 hr LT binning resolutions. Comparison of the numbers 40 and 9,216 shows clearly that the seasonal or day number of year variation does not have good input data for the entire African region. As presented previously, this might be the reason for the shortfall in NN TEC model to capture the EIA feature in some cases. In order to have fairly adequate data, we did not apply restriction to the horizontal smear. Therefore, in a day, there were about 80 observations as shown with red stars in Figure above. It is clear that the ~80 COSMIC TEC data points in a day are still far less compared to 9,216 data points needed to fill all the grid cells in a day. This problem was partly solved by

adopting appropriate data binning criteria. For example, instead of binning data according to year, we binned data according to only three different solar flux levels (as in Mungufeni et al, (2019), *Characterization of Total Electron Content over African region using Radio Occultation observations of COSMIC satellites*, Adv in Space Res 65, 19 – 29).

We established that the COSMIC TEC data values with smear > 1500 km do not introduce alarming errors. This was done by analyzing COSMIC TEC data which were coincident with TEC observed by ionosonde stations at Hermanus, Grahamstown, and Louisvale. The observations of the year 2013 were considered. Table below presents the root mean squared error between (i) ionosonde and COSMIC TEC without limiting the horizontal smear, and (ii) ionosonde and COSMIC TEC with horizontal smear limited to 1500 km.

| Station | Smear < 1500 km | | No limitation | |
|---|---|---|---|---|
| | Number of observations | RMSE (TECU) | Number of observations | RMSE (TECU) |
| Hermanus | 38 | 1.838 | 65 | 2.256 |
| Grahamstown | 34 | 6.479 | 73 | 7.923 |
| Louisvale | 42 | 2.765 | 91 | 3.252 |

The table shows that the RMSE for the two cases over a particular ionosonde station are not grossly different. Based on these results, trading off accuracy may not be costly compared to trading off adequate need of data. Therefore, we decided not to impose any restriction on the horizontal smear. Although the RMSE appear to be smaller when

the smear < 1500 km, some of the data points that were subjected to this restriction are also far from the linear least squares fitting line. See blue stars in Figure below.

[Figure]

**Comment:**

I strongly suggest to compare the output of their model with ionospheric TEC (up to 700 km) from all over four stations Digisonde stations over South Africa https://spaceweather.sansa.org.za/products-andservices/current-conditions/ionograms. This will provide a much more realistic comparison test to their model

**Response:**

*The suggested idea has been implemented in pages 19 - 21.*

We illustrate in Figure below with magenta lines the diurnal patterns of TEC measured by ionosonde stations at Hermanus (panels in column (i)), Grahamstown (panels in column (ii)) and Louisvale (panels in column (iii)). The corresponding TEC generated by our spline technique model (spline), Nequick 2, and IRI-2016 are superimposed with red, green and blue lines, respectively. We need to mention that during computation of TEC using NeQuick 2 and IRI-2016, the height was limited to the approximate altitude of the COSMIC satellites (800 km). The panels in rows (a) - (c) show TEC on day of year 170 (June), 260 (September), and 350 (December), respectively. All these three days of the

year 2013 were geomagnetically quiet. Preliminarily, figure below appears to reveal that IRI-2016 either overestimates (December) or underestimates (June and September) the TEC measured by the ionosonde stations. On the other hand, our spline model and NeQuick 2 seem to depict good correspondence between the observed and the modeled TEC. It can also be seen from figure below that over a particular station, the shape of curves on different days representing TEC generated by the IRI-2016 and NeQuick 2 models are similar. This is expected since these two models were meant to reproduce monthly median values of the ionosphere. This means that our model, based on spline functions may capture better the day-to-day variability of the ionosphere.

[Figure]

We generated such data plotted in the above figure for the entire year 2013 and then perform statistical analysis of the differences between the observed and the model TEC

data. Table below presents in columns 3 the correlation coefficients, r for the correlations between modeled and ionosonde TEC. Moreover, the table presents the RMSE when the ionosonde TEC was estimated using the models listed in column 2. The number of observations, n over each station that were used to determine, r and RMSE are put in brackets below the station name.

| Ionosonde Station /number of observations | Model | r | RMSE (TECU) |
|---|---|---|---|
| Hermanus (n = 5,110) | Spline | 0.92 | 4.64 |
| | IRI-2016 | 0.86 | 5.45 |
| | NeQuick 2 | 0.92 | 4.10 |
| Grahamstown (n = 4,450) | Spline | 0.88 | 5.56 |
| | IRI-2016 | 0.82 | 6.29 |
| | NeQuick 2 | 0.86 | 5.27 |
| Louisville (n = 4,543) | Spline | 0.94 | 3.82 |
| | IRI-2016 | 0.87 | 5.62 |
| | NeQuick 2 | 0.94 | 3.73 |

It can be seen from the Table that the r values associated with NeQuick 2 and spline based model are consistently better when compared with that of IRI-2016. Moreover, the RMSE values associated with IRI-2016 are the highest in all the cases. These two observations indicate that compared to spline and NeQuick 2, IRI-2016 poorly estimates TEC at the locations of the ionosondes. The RMSE values associated with NeQuick 2 are always slightly lower than that of spline, while the r values associated with spline are mostly comparable or slightly higher than that of NeQuick 2. These discussions demonstrate that our spline model generates TEC values consistently with that observed by ionosondes. This implies that equivalent TEC measured by ionosondes over locations which do not have ionosonde stations can be predicted fairly well using our model.

**Comment:**

The manuscript presents an empirical model describing ionosphere total electron content over African region. Authors use experimental TEC data obtained using dual frequency GNSS RO receivers onboard of COSMIC satellites to construct the model. They validate the model using same type of data that was used to construct the model but for a different period.

**Response:**

In addition to using the same type of data for validating our model, we have used TEC measured by ionosonde stations over South Africa. See pages 19 – 21 in the revised manuscript. Also see the preceding response to last comment of reviewer #1.

**Comment:**

General impression is that the present work has no contribution to the current understanding of the low latitude ionospheric physics/modelling. The work brings a little science and the newly created model could hardly be used in any real-life application. Authors are making too many assumptions and mistakes, sometimes trying to deliberately present performance results better than they are. Moreover, the performance of the model has not been compared to any other well-known model, leaving a room for doubts. Therefore, I recommend the manuscript (in its present form) is **rejected**. At the same time, the work might be improved and worth publication after substantial modifications. Please find below a list of critical issues along with possible improvements/corrections for a potential future re-submission.

**Response:**

Since all the comments by the editor and reviewers have been addressed appropriately, we do not expect a decision to reject our manuscript.

**Comment:**

P.1 L.27: Replace "good" with "applied". Otherwise, provide a proof of the model "goodness"

**Response**:

The suggestion has been implemented on page 1, line 22.

**Comment**:

P.2. L.35-38: Not all GNSS systems support ionospheric corrections. E.g. GLONASS does not broadcast any ionospheric model parameters. Correct the sentence accordingly.

**Response**:

The suggested correction can be seen on page 2, line 7 - 8.

**Comment**:

P.2 L.40: Provide a reference to the original description of Klobuchar model: *"Klobuchar JA (1987) Ionospheric time-delay algorithm for single frequency GPS users. IEEE Trans Aerosp Electron Syst 23(3):325–331. https://doi.org/10.1109/TAES.1987.310829"*

**Response**:

The suggested reference has been added on page 2, line 11 and page 28, lines 1 - 3.

**Comment**:

P.2 L.41-42: NeQuick G model is based on the NeQuick model, but not NeQuick 2. Correct the statement and the reference accordingly, e.g. *"EC (2016) European GNSS (Galileo) Open Service—Ionospheric correction algorithm for Galileo single frequency users, Issue 1.2, Sept. 2016, European Commission"*

**Response**:

The suggested correction can be seen on page 2, lines 12 - 15.

**Comment**:

P.2. L.42: Change "The NeQuick is" to "The NeQuick and its subsequent modifications (NeQuick G and     NeQuick 2) are"

**Response**:

The suggestion has been implemented on page 2, lines 15 - 16.

**Comment**:

P.2 L.53: IRI model does not provide information about "electron and ion velocities". It only provides information about equatorial vertical ion drift. Correct the sentence accordingly.

**Response**:

The sentence has been corrected as below.

" …….parameters (such as electron density, electron and ion temperatures, and equatorial vertical ion drift), ……"

In the revised manuscript, it appears as seen on page 2, lines 26 - 27.

**Comment**:

P.2 L.55-56: Change "The model is primarily" to "IRI is an empirical model primarily"

**Response**:

The suggestion has been implemented as seen on page 3, line 1.

**Comment**:

P.3 L.74: Change "GIM" to "global ionosphere model", as GIM is already defined to be Global Ionosphere   Map.

**Response**:

The suggestion has been implemented in page 3, line 19.

**Comment**:

P.3 L.76: Change "GIM model" to "global ionosphere model"

**Response**:

The suggestion has been implemented in page 3, line 21.

**Comment**:

P.3 L.80-82: The high values of RMS in low latitude region provided by CODE is,

primarily, due to the inability of the selected model function (spherical harmonics) to describe ionospheric structure in low latitude. Modify the sentence accordingly.

**Response**:

We based on Fig below (obtained from Najman, P. and Kos, T.: Performance Analysis of Empirical Ionosphere Models by Comparison with CODE Vertical TEC Maps, Chapter 13, in: Mitigation of Ionospheric Threats to GNSS: an Appraisal of the Scientific and Technological Outputs of the TRANSMIT Project, InTech Open Science publications, pp. 162 - 178, doi:10.5772/58774, 2014) to make the statement, "This could be due to the poor distribution of IGS tracking stations over Africa and anomalies in the ionosphere related to the geographic and geomagnetic location".

[Figure]

Indeed, figure above shows high RMS values over the oceans and land masses that have few/no ground based GPS receivers. This situation typically exists around and over the African continent.

Since figure above does not strictly show high values of RMS over all low latitude regions where EIA exists, we removed EIA as a reason for the high RMS values over

Africa. We have included "inability of spherical harmonics function …." on page 3, lines 26 – 27.

**Comment**:

P.3 L.84: Change "the GIM model" to "global models"

**Response**:

The suggestion has been implemented in page 3, line 29.

**Comment**:

P.4 L.115: Author use TEC integrated up to COSMIC satellite heights (800 km) to construct the model (*"integration being done up to the altitudes of the COSMIC satellites"*). However, the topside TEC values (according to numerous studies, e.g. by Bilitza 2009, Yizengaw 2008 etc.) can reach from 10% to 80% of the total electron content (from ground to GNSS satellite heights). This fact significantly reduces the scientific value and application of the developed model. Essentially, the model is useless for GNSS   applications.

**Response**:

This comment is similar to that of reviewer #1 to which we already responded in this document on pages 4 - 5. In the revised manuscript, we justified lack of inclusion of plasmaspheric TEC in COSMIC RO TEC in page 5, lines 12 - 29 and page 6, lines 1 - 5. The significance of the current study has been provided in page 6, lines 2 – 5 and page 1, lines 27 - 31.

The highlights of justifications of lack of inclusion of plasmaspheric TEC and significance of this study are as follows.

In order to get integrated electron density approximately up to the altitudes of GPS satellites, Okoh et al., (2019) used neural networks to learn the relationship between coincident TEC measurements done by ground based GPS receivers and COSMIC RO. They showed that the ratio between TEC data from the two sources vary spatially. This observation implies that the neural networks may not learn very well the relationship between TEC measured by ground-based GPS receivers and COSMIC RO over locations which do not have the former data set during the entire study period. As

shown previously in response to reviewer #1, there were large spatial coverage's that do not have ground based GPS receivers. Due to such problems, we used only COSMIC TEC without any adjustments.

We examined the available differences between coincident COSMIC RO TEC and ground based GPS TEC. Concerning the region under study, the upper quartile of the differences between coincident COSMIC RO TEC and ground based GPS TEC could reach ~11 TECU (Mungufeni et al, (2019), *Characterization of Total Electron Content over African region using Radio Occultation observations of COSMIC satellites*, Adv in Space Res 65, 19 – 29). Since the upper quartiles of the differences can reach up to ~11 TECU, the median/mean values might obviously be much lower than this value. This might be the reason for observing most of the well known ionospheric TEC features over the African region when the COSMIC RO TEC were appropriately binned in the above reference. The ionospheric features being referred to include; (i) occurrence of minimum and maximum TEC during 0:00–08:00 LT and 12:00–16:00 LT respectively, (ii) occurrence of secondary TEC enhancement (maximum) during 16:00–20:00 LT, (iii) lowest TEC values being observed in June solstice and highest TEC values observed in March equinox, (iv) TEC values increase as solar activity changes from low to high, (v) mid latitude TEC values are lower than those of low latitude regions, and (vi) occurrence of equatorial ionization anomaly.

Therefore, the current model was built with the aim of reproducing these known ionospheric features. Such endeavors are important for educational purposes. Moreover we illustrated in a previous response that our model generates TEC values consistently with that observed by ionosondes. This implies that equivalent TEC measured by ionosondes over locations which do not have ionosondes can be predicted fairly well using our model.

**Comment**:

P.5 L.124-126: This statement "*Since the magnitudes of the TEC obtained from COSMIC occultation 124 measurements are close to ground based GNSS TEC*", is not consistent with the previous statement and studies by Mungufeni et al. 2019. Where

they show that, depending on the location, the RMS error can vary from 2 to 8 TECU and error distribution plots show values from -24 to 20 TECU. Such large errors cannot be considered "*close to ground-based GNSS TEC*". Authors, at least, are expected to provide information about relative TEC errors (in %, rather than TECU) to claim that errors can be tolerated (if so).

**Response**:

In addition to the quoted sentence which appears in the revised manuscript on page 5, lines 23 – 27, on page 5, lines 27 – 29 and page 6, lines 1 - 5, we have added the following information.

Since the upper quartiles of the differences can reach up to ~11 TECU, the median/mean values might obviously be much lower than this value. This might be the reason for observing most of the well known ionospheric TEC features over the African region when the COSMIC RO TEC were appropriately binned in the above reference. The ionospheric features being referred to include; (i) occurrence of minimum and maximum TEC during 0:00–08:00 LT and 12:00–16:00 LT respectively, (ii) occurrence of secondary TEC enhancement (maximum) during 16:00–20:00 LT, (iii)   lowest   TEC values being observed in June solstice and highest TEC values observed in March equinox, (iv) TEC values increase as solar activity changes from low to high, (v) mid latitude TEC values are lower than those of low latitude regions, and (vi) occurrence of equatorial ionization anomaly.

Therefore, the current model was built with the aim of reproducing these known ionospheric features. Such endeavors are important for educational purposes. Moreover we illustrated in a previous response that our model generates TEC values consistently with that observed by ionosondes. This implies that equivalent TEC measured by ionosondes over locations which do not have ionosondes can be predicted fairly well using our model.

**Comment**:

P.6 L.150: The title of the reference Emmert et al. 2010 is incorrect: *Emmert, J. T., Richmond, A. D., and Drob, D. P.: Statistical analysis of the correlation 412 between the equatorial electrojet and the occurrence of the equatorial ionisation 413 anomaly over the East African sector, J. Geophys. Res., 15; A08322; 414 doi:10.1029/2010JA015326, 2010.*

**Response**:

The correction has been made. See page 27, lines 2 – 4.

**Comment**:

P.6 L.157-167: The selected spatial resolution of 15º in longitude and 5-8º in latitude is too coarse to describe the ionosphere reasonably, especially for the low latitude region, where TEC is changing dramatically from the crest down/up to two peaks of EIA. E.g. GIM maps (the source of the data for most of the empirical models discussed by the authors in the introductions section) use at least 5º by 2.5º resolution (lon and lat). Moreover, 15º in longitude corresponds to 1 hour in LT. Gradients in TEC as a function of LT during sunrise and sunset hours may reach tens of TECU per hour (e.g. Mungufeni et al. 2019, Fig. 2). Therefore, such coarse spatial resolution in longitude will lead to big errors in the model description.

**Response**:

We have now developed the model using data binned in grids with longitude range of 5º and latitude range of 3º. See page 7, lines 10 – 12 and page 12, lines 4 - 6. This fairly high resolution allowed us to observe the EIA features as seen in Figure 7 on page 24.

**Comment**:

P.6 L.170: The whole solar cycle 24 has relatively low solar activity level compared to the two previous ones. Nevertheless, even if we look only at the 24th solar cycle, 2011 and 2016 could hardly be attributed as years of high solar activity level. Please, modify the statement accordingly (e.g. as it is done on P.7 L.182).

**Response**:

The sentence has been modified as shown on page 7, line 1.

**Comment**:

P.7 L.189: Please clarify, how 36 solar flux bins were obtained. From the description, it is only 3 solar flux ranges and 12 months, that gives 36 (3x12). But when listing by a variable, only number 3 has to be specified, as it is done, for example with the rest of the variable (hour, lat and lon). Indeed, if we take 60,480 TEC values indicated in L.189, this number can be obtained by multiplying 5x14x3x12x24, but not 5x14x36x12x24.

**Response**:

In each of the 3 solar flux ranges, there are 12 nodes, corresponding to the months in a year (see table 2). This results into the 36 solar flux nodes. The listing in equation 1 on page 11 has reflected this. Moreover the longitude and latitude nodes have changed to 16 and 24, respectively.

**Comment**:

P.8 L.205: According to the definition of cubic spline, it is a spline constructed of piecewise third order polynomials, meaning none of the B splines used in the model were cubic (order 2 and 4). Change the "cubic B spline" into "B spline of different orders" throughout the text and abstract.

**Response**:

The suggestion has been implemented. See page 11, lines 24 - 25; page 1, line 17; and page 25, line 9.

**Comment**:

P.9 L.218-220: Consider changing this sentence to something like "In order to assess the ability of the model to describe the data used to construct the model, modelled data were compared to the experimental one. The results of the self-consistency check are presented in Figure 1."

**Response**:

This suggestion has been implemented in the manuscript on page 12, lines 10 – 12.

**Comment**:

P.9 L.228-229: It is surprising that the authors compare the results of the climatological model (i.e. model where input data were averaged over time, e.g. one month) with GIM map for a single day of that month. Such a comparison is not correct. On top of that, by

looking at TEC maps obtained from COSMIC and later by B spline model (columns 2 and 1), one can hardly see any separation between the peaks of the EIA, that can, taking into account averaging in all the bins (e.g. lat and lon) performed by authors, hardly be comprehended.

**Response**:

Now the GIM-TEC panel has been removed. See figure 1 in page 13 (see also figure below). After reducing the spatial resolution of the data, figure 1 showed distinct two crests and a trough, particularly before 18:00 LT. After this time, when the zonal electric field reverses westwards, upward plasma drift (responsible for EIA formation) is altered, leading to no separation of crests.

**Comment**:

P.9 L.231-232: By looking at the color plots, a reader can hardly assess the performance of the model. It is suggested, in addition to the plots, to present/discuss the results of the mismodelling in terms of a bias and RMS of the error.

**Response**:

Third column panels in Figure 1 on page 13 (see also figure below) present the error map which shows error values < 0.1 TECU.

[Figure]

These really small magnitudes of the errors may not require statistical analysis to demonstrate that the errors can be tolerated.

**Comment**:

P.10 L.250-252: From Fig 1 it cannot be clearly understood the secondary maximum if any, especially at -20 lat. Please, if you discuss a feature, try demonstrating it clearly to the reader. A separate figure, or at least, a dashed line at -20 and 4 in Fig 1 is needed to support the statement.

**Response**:

The current figure 1 on page 13 (also see above figure) has a good mark for pointing at the two peaks of diurnal TEC. In revised manuscript, see text on page 13, lines 14 – 15. The mark used to point at the two peaks is the magenta line on panels in row (c). We computed at lon 37.5º N, the corresponding geographic latitude of the magnetic equator as ~9º N. Figure above shows that the trough is not centered on the magnetic equator. Past studies (e.g Mungufeni et al., (2018): Statistical analysis of the correlation between the equatorial electrojet and the occurrence of the equatorial ionization anomaly over

the East African sector, Ann. Geophys., 36, pp. 841 – 853, 2018) show that the EIA trough is not exactly centered at the magnetic equator, particularly at the location of figure above.

**Comment**:

P.11 L.269-270: In row (b), Fig 1, none of the panel show peaks of the EIA. There is no clear separation of the crest and peaks of EIA. Nor in panels b1/b2 neither in b3. Modify the sentence accordingly.

**Response**:

This comment about separation of EIA crests was previously raised and we already responded.

**Comment**:

P.277-279: The structure of the crest might differ based on various factors (including level of the geomagnetic disturbance). However, when taken as an average, a clear 2 peak structure is present in low latitudes, representing EIA.

**Response**:

Based on the remark in the comment we have removed the statement about observation of several crests and discussions associated with it.

**Comment**:

P.12 L.298-299: The science question in this case is not how to model the observed data, but how to explain the data. What is the physical explanation for the absence of the EIA structure (two peaks and the crest) in TEC values calculated from the ground up to COSMIC satellite heights (~800km).   And whether this phenomena is not a limitation of the technique applied to calculate TEC. Namely,    TEC computed by integrating electron density profile, that by itself is a product of RO inversion, is subject to big errors, especially in places where big horizontal gradients exist (read, e.g. M.M Shaikh et al., Implementation of Ionospheric Asymmetry Index in TRANSMIT Prototype, DOI:    10.5772/58551). Without understanding the reasons of the observed behavior all the modeling efforts are meaningless.

**Response**:

In the paragraph under question, we mentioned asymmetry of EIA feature and occurrence of secondary peak in TEC over Africa. We further mentioned that these features can be seen in the data we used to develop our model. Therefore, our model emulates these features. We would like to mention that these two features have been well explained in the manuscript (see page 13, lines 12 – 18, page 14, lines 1 – 9; and page 16, lines 3 – 9.

The reviewer's phrase, "absence of the EIA structure (two peaks and the crest) in TEC values calculated from the ground up to COSMIC satellite heights (~800km)" did not exist in our manuscript. This makes it difficult for us to understand the point the reviewer would like to make. Anyway, we guess that the reviewer is talking about the absence of asymmetry of EIA feature in GIM-TEC. In case this is correct, as stated before, we removed presentation and discussions of comparison of our model with GIM-TEC.

**Comment**:

P.13 L.313: One cannot see the "perfect match" of the observed and modelled data just by looking at the plots. At least a third row in form of difference map (error map) has to be presented to visually    assess the error level. Moreover, statistical results (e.g. RMS and bias of the error) must be presented in    order to make such a bold conclusion.

**Response**:

We have provided in the third column panels of figure below (see in revised manuscript Figure 1 on page 13) the error levels between the observed and the measured data.

[Figure]

The error maps show values < 0.1 TECU (see the color bar). This means as expected, the observed and the modeled data almost match perfectly. Without performing statistical analysis, the fact that all error values are < 0.1 TECU gives the confidence of making the conclusion that the observed and modeled data almost match perfectly. Performing statistical analysis would be necessary if some error values were fairly high.

**Comment:**

P.13 L.312-324: Authors do not discuss at all the TEC behavior observed in September at lat ~ -20, where its diurnal variation has a maximum during local night hours (21-03 LT). This maximum seems to exceed any other TEC values on this plot (row c, column 1 and 2) and looks like an error in the data processing.        Such behavior seems to have no physical explanation.

**Response**:

After carefully analyzing the data again the feature mentioned in the comment was no longer seen. In the revised manuscript, see figure 2 on page 15. The figure is reproduced below.

[Figure]

**Comment:**

P.14 Section 5: The authors fail to explain why they need yet another TEC model. Unless the performance of the newly created model is compared to existing models and it is demonstrated that its any better than the rest of the models present on the "ionosphere model market" (e.g. IRI, NeQuick,  NTCM etc.), there is very little value in the study (both scientifically and application-wise).

**Response**:

As seen in our previous responses to reviewer #1 in this document on page 13, we compared our model with the existing models such as IRI, NeQuick, and AfriTEC (Okoh et al. 2019). Also see in the revised manuscript sections 5.2 and 5.3. In page 20 of this document, we had provided the scientific significance of this study. On page 19 of this

document, we provided the page and line numbers in the revised manuscript where the scientific significance of this study can be seen.

**Comment:**

P.15 L.350-353: Figure 4 does not show the full picture of the error distribution. It is clearly cut at -14 and 14 TECU. Iif one looks at Figure 3, errors in TEC can easily reach +-20 TECU (just draw a vertical line at any value of Observed TEC, e.g. at 30 TECU). It looks like the authors deliberately try to improve the results of their model performance.

**Response:**

Actually, we should have not indicated ±16 on the horizontal axis. Moreover, we should have indicated on the horizontal axis < -14 (instead of merely -14) and > 14 (instead of merely 14). The total number of errors with values in the range of -14 – 14 TECU was 16858 (97.4 %), while the number of errors with values outside this range was 454 (2.6 %). By comparing these two percentages, it can be deduced that the number of errors with values outside the range of -14 – 14 TECU was insignificant. After implementing the above changes, Figure 4 would appear as below. In the revised manuscript, see on page 18. We have also superimposed the percentages of the different error levels. This is in accordance with the previous comment of the reviewer.

[Figure]

**Minor/Typo comments:**

**Comment**: P.1 L.17: Change "derived" to "obtained"

**Response**: Done as seen on page 1, line 13

**Comment**: P.1 L.19: Change "Geomagnetically quiet time (Kp < 3 and Dst > -20 nT) data during the years" to "Data    during geomagnetically quiet time (Kp < 3 and Dst > -20 nT) for the years"

**Response**: Done as seen on page 1, line 15 - 16.

**Comment**: P.1 L.22 Change "to obtain the model" to "to obtain model coefficients"

**Response**: Done as seen on page 1, line 18.

**Comment**: P.1 L.26 Change "COSMIC TEC" to "COSMIC RO TEC"

**Response**: The sentence associated with the phrase was stating that our model is the first over the entire African region. After the publication of Okoh et al., (2019), this became invalid. Therefore, the sentence that contained the phrase in the comment was removed.

**Comment**: P.2 L.31: Change "using Global Navigation Satellite Systems" to "in Global Navigation Satellite Systems"

**Response**: Done as seen page 2, line 2 - 3.

**Comment**: P.2 L.30 Change "during day" to "during the day"

**Response**: The corrected can be seen on page 2, line 11.

**Comment**: P.2. L.49: Space is missing between "European Geostationary"

**Response**: See page 2, line 23.

**Comment**: P.2 L.50: Change "GPS And Geo-Augmented Navigation" to "GPS-aided Geo Augmented Navigation"

**Response**: see page 2, line 24.

**Comment**: P.3 L.63: Space is missing in "analysis centers"

**Response**: See page 3, line 11 - 12.

**Comment**: P.3 L.64: Space is missing in "using the"

**Response**: See page 3, line 10.

**Comment**: P.3 L.64: Change "Global Ionospheric TEC data Map (GIM)" to "Global Ionosphere Maps (GIMs)   containing vertical TEC data"

**Response**: See page 3, line 9.

**Comment**: P.3 L.66: Change "Global Ionospheric TEC data Maps (GIMs)" to "GIMs". It has been defined two lines above.

**Response**: See page 3, line 11.

**Comment**: P.3 L.70: Space is missing in "the average"

**Response**:   See page 3, line 14 - 15.

**Comment**: P.3 L.71: Space is missing in "by CODE"

**Response**: See page 3, line 16.

**Comment**: P.3 L.76: Space is missing in "constructed a"

**Response**: See page 3, line 21.

**Comment**: P.3 L.77: Space is missing in "GPS radio"

**Response**: Page 3, line  22.

**Comment**: P.3 L.82: Space is missing in "related to"

**Response:** The phrase was contained in a sentence which stated that the high RMSE values were due to EIA. After observing that some EIA regions depicted low RMSE values, the statement became invalid. Therefore, the sentence which contained the phrase in the comment was removed.

**Comment**: P.4 L.87: Change "localized ionospheric structure" to "localized ionospheric structures"

**Response**: See page 4, lines 3 – 4.

**Comment**: P.4 L.88: Change "on a global scale model" to "in global models"

**Response**: See page 4, line 4.

**Comment**:P.5 L.140: Space is missing in "during geomagnetically"

**Response**: See page 6, lines 18 - 19 .

**Comment**:P.6 L.147: Change "solar activity" to "solar activity level"

**Response**: See page 6, line 26.

**Comment**:P.6 L.164: Remove "15" in "reduced 15 to 5"

**Response**: Since the spatial resolutions were changed, the phrase in the comment was removed.

**Comment**:P.7 L.181 Space is missing in "the F10.7"

**Response**: See page 9, line 1.

**Comment**:P.9 L.223: Change "Global Ionosphere Map (GIM) TEC (GIM-TEC)" to "GIM TEC", as it was defined     earlier, remove "Center for Orbit Determination in Europe" – it was defined earlier

**Response:** Since the reviewer did not recommend comparison of our model with the CODE GIM, the phrase in the comment was removed.

**Comment**:P.9 L.225-226: Remove "The daily GIM-TEC values are derived using the GNSS data collected  from over 200 tracking stations of IGS and other institutions", as this information was given earlier in the text

**Response**: For the same reason given in the preceding response, the phrase in the comment was removed.

**Comment**:P.10 L.238: Space is missing in "in turn"

**Response**: See page 12, line 27.

**Comment**:P.14 L.336: Change ";" to ":"

**Response**: See page 16, line 22.

**Comment**: P.14 L.337: Space is missing in "root mean squared"

**Response**: See page 16, line 23.

**Comment**: P.17 L.373: Change ":" to "." In "0.93"

**Response**: See page 25, line 14.

[revised manuscript text omitted]

---

## Referee Report (RR1)

Angeo-2019-160

Reviewer #3

Comments on the paper:

Modeling Total Electron Content derived from radio occultation measurements by COSMIC satellites over the African region

Reference file: angeo-2019-160-manuscript-version4.pdf

The authors present a regional modeling of the TEC deduced from the vertical profiles of the ionospheric density, obtained by radio occultation with the COSMIC satellites over Africa. Despite using the entire database spanning a decade (2008-2018) and limiting themselves to magnetically calm days, they do not have enough measurements to fit these measurements to different variables in their model. First, they present their interpolation algorithm to have a value at each point of a geometric grid. The result of their model is discussed on our knowledge of variations in the ionosphere.
It is difficult to comment on an article which has already been extensively modified following numerous comments from the 2 previous referees. However, I still have a list of remarks that I have positioned in minor (m) and major (M):

(m) Paragraph 'Introduction' on 3 pages (p2-4).
The first 24 lines present the ionospheric models used in single frequency for the GPS and Galileo systems. The following 7 lines relate to the IRI model. The following 16 lines talk about GIM maps obtained by processing GNSS measurements. Finally, on line 31, p.3, the word COSMIC appears to present a global model of GNSS and RO measurements and to conclude that the result over Africa is different from CODE's GIM maps.
That's a lot of text that takes a bit away from the topic being discussed. I would have preferred a more direct introduction based on the 3 keywords of the title:
- Why did you choose an area above Africa where additional measurements are and when a choice above Europe or North America would have made it possible to use the many vertical ionosondes to validate the model?
- Why use only RO / COSMIC measurements to build a model and which constitutes the originality of this work?
- Why a mathematical model with spline functions? Currently, the word "spline" occurs only once, in the sentence on line 24, p4.
[It is likely that much of the current text of the introduction would have been found as a result of this questioning].

(m) p.5, line 9. The given database contains ionization profiles. To obtain VTEC, you must integrate up to the altitude of the COSMIC satellites. This altitude (~ 800 km) is not always specified (or later in the text) but it is important for a future comparison with other VTECs. It

is certain that all the profiles do not give the same final altitude and then two questions arise:
- if a profile stops at 600 km for example, what do the authors with this measurement?
- and more generally, do the authors make a selection and if so, on what criteria?

(m) p.9, lines 6-12. Following the comment on referee # 2, the text has been modified to understand that the 36 solar variables are indeed the 3 levels (L, M, H) repeated each month (3 * 12 = 36, table 2). The rationale is '**to represent seasonal TEC**'. But then, why keep a variable "months" (to study seasonal) which will require a multiplication of the number of variables by 12 (line 11)?

(m) p.9, table 2. I have not read the information, but I assume that the monthly flow values in table 2 relate only to the dates of the measurements used in the model. However, in 2012, we are in strong solar activity of SC#24 and perhaps with values of flux higher than those present. In this case, the validation will relate to an extrapolation, which can quickly lead to important TEC values?

(M) p.10, line 7. When there is little (or no) value in a box, the authors adopt a smoothing by splines. There are a lot of spline functions and the authors don't specify their choice. For cubic splines, one can obtain oscillations with stronger extreme values since the smoothing passes through the measurement points. For other spline functions, there is a reduction in variability since the interpolated curve passes between the points. The approach followed by the authors is important for the rest of the work and I think it would have been interesting to illustrate some typical cases with figures.
When there are multiple points in a box, the authors take the mean value? What is the variability (min / max) which gives information on the uncertainty of the measurement?
When a node has a sufficient number of measurements, do the authors keep the average or opt for the interpolated value?
Only one mathematical reference (deBoor, 1978) in this article is, in my opinion, too weak (compared to 38 geophysical references in the bibliography).

(m) Pages 10-11. The authors propose a 3-step algorithm for filling the geometric grid of measurements. It's a bit of an empirical method. Have the authors analyzed other interpolation methods starting from an irregular grid?

(m) p.11, line 15. I did not understand the convergence of the procedure after 3 rotations. Need to iterate until all the boxes are filled and maybe a number of 3 is not enough? At this stage, I think that the authors could have presented TEC histograms on 3,981,312 bins against the 121,447 bins input. Is it the same distribution (mean, rms)?

(m) p.13, line 13. The authors justify the quality of their model by the existence of a secondary peak at the magnetic equator already observed elsewhere. However, if I make a vertical line around 16 LT for example on the observed or on the model (Figure 1), I will see an irregular variation of the TEC (~ 10 tecu) in latitude with many secondary peaks (southern hemisphere for example) and not a steady decrease as expected. These secondary peaks are not physical and are due to averaging (hence my question about variability in a cell) and to interpolation. What do the authors think?

(m) p.14, line 11-20. I do not see quite the same thing that the authors describe in particular for the graph c in strong solar activity. Maximum north is on the equator (the 2 bubbles red colored) when expected at 20 ° N? The south EIA maximum appears to be well positioned.

(m) p.16, line 10. A first evaluation is made only on longitude 37.5 ° E due to the existence of GPS measurements and publications around these measurements. Fortunately, there are African stations at other longitudes. My question: since the model is built between -20 and 60 °E longitude, are the conclusions of 37.5 °E longitude valid for other longitudes? I would have seen an overall statistical result but I have no idea because the difference (observed-modeled) is less than 0.1 tecu on the 2 examples. Is this same conclusion for all longitudes?

(m) p.18, line 10. The authors do not provide any positioning on the 1600 points with an absolute difference modeling of at least 10 tecu. It is certainly for the year 2012 and not 2018 but the points relate to a particular hour or month? [I already pointed out a possible divergence of the model in one of my previous remarks in the case of an extrapolation with solar activity].

(M) p.21, lines 14-16. The authors validated their model with ionosondes in South Africa, therefore located in mid-latitudes. I think it is an exaggeration to say that we would have the same result ('predicted **fairly well** using our model.') With a low latitude ionosonde, the study remains to be done!

(M) p.25, line 4. I do not agree with this conclusion. It's because the TEC variations are more irregular with the spline model compared to the NN model that it is the best! Admittedly, the variations of TEC with NN are over-smoothed (but GIM / CODG also for example) but the many variations in Figure 7 are first linked to the error on the profile estimated by RO and by the procedures of interpolation to give values the nodes of the grid which is a **mathematical** filling and not a **physical** one.
I also regret that the comparison of Figures 6 and 7 is purely visual and that there are no statistical figures of differences in the proposed text.

My conclusion is that there is a **real work of exploiting the RO data** for modeling purposes. The initial difficulty is the lack of measures to cover the Africa zone. Also, the authors were forced to introduce mathematical approaches to cover all the variables retained. They justified their model on a physical result of maps reproducing the large known variability's. The model does not allow a fine-grained approach to the ionosphere compared to a more regional modeling with GNSS measurements. Their current conclusion is that their model leads to better results than the 2 empirical models (IRI and NeQuick) widely used. If the authors want to see their results applied to future studies, they must publish the coefficients of their model. Is this an objective of the authors?

---

## Author Response (AR2)

[revised manuscript text omitted]
" has undergone the second revision. Although the manuscript still needs additional corrections and clarification of some specific points raised by opponents, I am pleased to inform you that the current status of your article is a "minor revision".

**Response:**

We thank the editor and reviewers for taking time to evaluate our manuscript for the second time. All the comments in the two referee reports are addressed as shown below.

**Report #1**

**Comment:**

Authors need to mention which topside option they used when estimating IRI-16 TEC.

Response:

NeQuick model option was specified during estimation of topside electron density values. This information has been included in the current version of the manuscript. See page 21, lines 17 – 18.

**Comment:**

There are minor expression mistakes which should be corrected.

Response:

Since the reviewer did not specify the expression or provide examples of the expressions which have mistakes, it is difficult for us to interpret and understand his/her comment. As far as the manuscript is concerned, we do not see any mistake in equation 1 (page 13). There were few sentences which contained numerical values preceded by symbols > and <. In the current version of the manuscript, in all cases, spaces have been created between the symbol and the numerical value.

**Comment:**

Authors claim (Page 24 Line 7) that from Figure 7 clearly the position of the EIA trough can be identified. This is not very obvious in my view.

Response:

We need to mention that the statement (see page 26, lines 4 – 6) which has been copied and pasted below aimed at emphasizing existence of EIA trough, but not to pin point its position (central point) or width.

"Unlike our TEC maps in Figure 8 which clearly show the EIA trough (see magenta arrows) in all the seasons, the neural network technique TEC maps (Okoh et al., 2019) of Figure 7 only clearly capture the EIA trough in December solstice".

**Report #2**

**Comment**:
The authors present a regional modeling of the TEC deduced from the vertical profiles of the ionospheric density, obtained by radio occultation with the COSMIC satellites over Africa. Despite using the entire database spanning a decade (2008-2018) and limiting themselves to magnetically calm days, they do not have enough measurements

to fit these measurements to different variables in their model. First, they present their interpolation algorithm to have a value at each point of a geometric grid. The result of their model is discussed on our knowledge of variations in the ionosphere. It is difficult to comment on an article which has already been extensively modified following numerous comments from the 2 previous referees. However, I still have a list of remarks that I have positioned in minor (m) and major (M):

Response:

We are thankful to the reviewer for recognizing the enormous work done in this manuscript.

**Comment**:

(m) Paragraph 'Introduction' on 3 pages (p2-4). The first 24 lines present the ionospheric models used in single frequency for the GPS and Galileo systems. The following 7 lines relate to the IRI model. The following 16 lines talk about GIM maps obtained by processing GNSS measurements. Finally, on line 31, p.3, the word COSMIC appears to present a global model of GNSS and RO measurements and to conclude that the result over Africa is different from CODE's GIM maps. That's a lot of text that takes a bit away from the topic being discussed. I would have preferred a more direct introduction based on the 3 keywords of the title: - Why did you choose an area above Africa where additional measurements are and when a choice above Europe or North America would have made it possible to use the many vertical ionosondes to validate the model? - Why use only RO / COSMIC measurements to build a model and which constitutes the originality of this work? - Why a mathematical model with spline functions? Currently, the word "spline" occurs only     once, in the sentence on line 24, p.4. [It is likely that much of the current text of the introduction would have been found as a result of this questioning].

Response:

The answer to question 1: "Why did you choose an area above Africa where additional measurements are ……" can be found in the statement on page 4, lines 21 – 24. i.e,

"Due to the lack of a dense network of ground-based GNSS receivers and poor coverage of COSMIC RO data over the African region, the TEC model over the entire African region presented by Okoh et al. (2019) sometimes failed to capture the equatorial ionization anomaly (EIA) over the region."

Although the 2nd question: "why use only RO/COSMIC measurements …." does not have an answer/explanation in section 1, there is a link in the section (see sentence on page 4, line 26 - 27) to the statement which provides the answer/explanation in subsection 2.2 on page 8, lines 12 – 14. To minimize changes to the current version of the manuscript, we preferred to leave the statement in the same subsection. The statement being referred to is "Another reason might be the discrepancy which arises due to some locations being represented by adjusted COSMIC RO TEC while others by the ground-based GPS TEC data."

Answer to question 3 (Why a mathematical model with spline functions?) has now been provided as (see page 4, line 28 –page line 2): "These basis functions never vanish over limited intervals and add up to one at all local times and longitudes (De-Boor, 1978). Moreover, according to Scherliess and Fejer, (1999), they are ideally suited to model the equatorial ionosphere which exhibit smooth and rapid changes during daytime and near sunset, respectively, by proper placement of the mesh of nodes."

**Comment**:

(m) p.5, line 9. The given database contains ionization profiles. To obtain VTEC, you must integrate up to the altitude of the COSMIC satellites. This altitude (~ 800 km) is not always specified (or later in the text) but it is important for a future comparison with other VTECs. It is certain that all the profiles do not give the same final altitude and then two questions arise: - if a profile stops at 600 km for example, what do the authors with this measurement?  - and more generally, do the authors make a selection and if so, on what criteria?

Response:

The text on page 8, lines 6 – 8 and 20 – 29 demonstrate that we acknowledge existence of errors in COSMIC RO TEC. Ultimately; the text justified the usage of all available COSMIC RO TEC data. The texts on page 8 are as follows:

"Several studies (e.g. Krankowski et al., 2011 and Mengist et al., 2019) that have used COSMIC data commonly consider measurements with horizontal smear > 1500 km prone to errors and they reject such measurements."

"Although not presented here, we observed that the COSMIC TEC data values with smear > 1500 km did not introduce alarming errors. This observation was made when we analyzed COSMIC TEC data which were coincident with TEC observed by ionosonde stations over South Africa (see details in section 5.2) located at Hermanus, Grahamstown, and Louisvale. Interestingly, compared to measurements with horizontal smear > 1500 km, some measurements with horizontal smear < 1500 km were observed to be far from the linear least squares fitting line. Further analysis of COSMIC RO observations over our study area revealed that without restricting horizontal smear, there were ~80 RO measurements per day during the year 2013 (not shown here)."

 In the previous rebuttal, we demonstrated that measurements with small (<1500 km) horizontal smear may even be far away from the linear least squares fitting line compared to those with large (>1500 km) horizontal smear. This unexplained observation might be due to height profile <600 km, though this is not yet verified.

**Comment**:

(m) p.9, lines 6-12. Following the comment on referee # 2, the text has been modified to understand that the 36 solar variables are indeed the 3 levels (L, M, H) repeated each month (3 * 12 = 36, table 2). The rationale is '**to represent seasonal TEC**'. But then, why keep a variable "months" (to study seasonal) which will require a multiplication of the number of variables by 12 (line 11)?

Response:

The 12 DOY values (on the 15th of every month) correspond to the 12 months of the year. Therefore, the word "monthly" has remained. See page 10, line 7. Though seasons change with DOY, normally, the number of seasons in a year is < 12.

After examining the scripts again, it was realized that considering the solar flux levels L (F10.7 < 76 sfu), M (76 ≤ F10.7 ≤ 108 ), and H (F10.7 > 108 sfu) as separately having 12 values (totaling to 36 sfu values) was wrong. The correct procedure is to consider the solar flux levels L, M, and H as single values (totaling 3), where a value in a specific solar flux level in turn depends on the month as shown in Table 2.

**Comment**:

(m) p.9, table 2. I have not read the information, but I assume that the monthly flow values in table 2 relate only to the dates of the measurements used in the model. However, in 2012, we are in strong solar activity of SC#24 and perhaps with values of flux higher than those present. In this case, the validation will relate to an extrapolation, which can quickly lead to   important TEC values?

Response:

It is true that the solar flux values in Table 2 relate only to dates when there were measurements during the years 2008 - 2011 and 2013 – 2017. Since solar flux values vary daily, it is possible that some of the solar flux values in the year 2012 may be higher than those listed in Table 2 or solar flux values in the year 2018 may be lower than those listed in Table 2. In such a situation, the validation may lead to extrapolation, rather than interpolation. One of the limitations of B spline model is its inability to extrapolate. Therefore, in a scenario where the flux on validation day is higher (lower) than those listed in Table 2, the maximum (minimum) value listed in the table was considered. This idea was applied to day number of year (DOY < 15 and DOY > 350), longitude (lon < -17.5 and lon > 57.5º) and latitude (lat < -34.5º and lat > 34.5º) variations whose intervals are specified in section 3.

The above discussions are presented in the current version of the manuscript on page 18, lines 13 – 18.

**Comment**:

(M) p.10, line 7. When there is little (or no) value in a box, the authors adopt a smoothing by splines. There are a lot of spline functions and the authors don't specify their choice. For cubic splines, one can obtain oscillations with stronger extreme values since the smoothing passes through the measurement points. For other spline functions, there is a reduction in variability since the interpolated curve passes between the points. The approach followed by the authors is important for the rest of the work and I think it would have been interesting to illustrate some typical cases with figures.

Response:

The information that is required in the comment has been provided on page 10, lines 20 – 21 and 23 – 24, page 11, lines 6, 13, 17, 29, page 12, lines 1 – 16, page 13, lines 1 – 2.

In this study, smoothing spline and piece-wise cubic interpolation methods were used to estimate missing TEC data. As mentioned by the reviewer, the former method leads to reduction in data variability since the interpolated curve passes between data points, while the latter leads to the interpolated curve passing at the data points. Therefore, when slow variation of TEC was expected (e.g. after mid night, till about 10 a.m.) and there were at least a few (>2) TEC data available, smoothing spline data fitting method was used to estimate missing TEC values. In cases where rapid TEC variations are expected (e.g. during daytime, till local midnight) and at least half of the total expected number of data points were filled with TEC data, piece-wise cubic interpolation data fitting method was used to estimate missing TEC values. After estimating all missing data values, the diurnal TEC at spatial grid cells were then separately fitted with smoothing splines which were evaluated to obtain the TEC data that were later used to determine the model coefficients. Figure below demonstrates the appropriateness of our estimation of missing TEC data values and the use of estimated TEC data to determine model coefficients. In the figure, panels (a) – (c) present the available (*) and estimated (red line) TEC data values during low, medium, and high solar flux levels, respectively.

[Figure]

The TEC data plotted in above figure correspond to January and grid cell centered at longitude 17.5º W and latitude 34.5º S. The figure clearly shows that the available and estimated TEC data variations depict the well-known diurnal and solar activity level dependence patterns. Moreover, the figure shows that the available data values are in most cases close to the estimated TEC data values. Therefore, the estimated TEC data were then used to obtain the model coefficients.

**Comment**:

When there are multiple points in a box, the authors take the mean value? What is the variability (min / max) which gives information on the uncertainty of the measurement?

Response:

As added on page 10, lines 9 – 12, the average of the standard deviations of the bins that contained more than 1 TEC data during low (sample size = 21,108), medium (sample size = 6,180) and high (sample size = 7,495) solar flux levels were 1.28, 2.15, and 4.31 TECU, respectively.

**Comment**:

When a node has a sufficient number of measurements, do the authors keep the average or opt for the interpolated value?

Response:

During interpolation iterations, the available data were not replaced by interpolated values. As stated on page 11, line 29 and page 12, lines 1 - 2, after filling all missing data, the entire (both available and filled) diurnal data at a particular grid cell were then fitted with smoothing spline which were evaluated to yield the final data used to determined the model coefficients. This last procedure where available data were replaced by interpolated values might minimize the effects of outliers in the data.

**Comment**:

Only one mathematical reference (deBoor, 1978) in this article is, in my opinion, too weak (compared to 38 geophysical references in the bibliography).

Response:

All the ideas about spline interpolation and modeling were generated based on the one mathematical reference provided. We think that the information provided by the reference may not be doubted. Other mathematical information in the manuscript are common knowledge which may not need reference, like determining standard deviations, root mean squared error, and correlation coefficients.

**Comment**:

(m) Pages 10-11. The authors propose a 3-step algorithm for filling the geometric grid of

measurements. It's a bit of an empirical method. Have the authors analyzed other interpolation methods starting from an irregular grid?

Response:

We do not understand the idea of interpolation method starting from an irregular grid. Therefore, we did not try it.

**Comment**:

(m) p.11, line 15. I did not understand the convergence of the procedure after 3 rotations. Need to iterate until all the boxes are filled and maybe a number of 3 is not enough? At this stage, I think that the authors could have presented TEC histograms on 3,981,312 bins against    the 121,447 bins input. Is it the same distribution (mean, rms)?

Response:

It is true that iterations are supposed to run until all missing values are filled. For the data used in the current study, at the end of 3$^{rd}$ iteration all missing values were filled. It is important to remember that in an iteration there are 3 steps (sub-iterations).

As discussed in one of the previous responses in this document, considering the solar flux levels L, M, and H as separately having 12 values (totaling to 36 sfu values) was wrong and the correct procedure is to consider the solar flux levels L, M, and H as single values (totaling 3). Therefore, the total number of bins to be filled was 331776 (16 longitudinal, 24 latitudinal, 3 solar flux, 12 monthly, and 24 hourly bins), but not 3,981,312 (16 longitudinal, 24 latitudinal, 36 solar flux, 12 monthly, and 24 hourly bins). These discussions are reflected in the manuscript on page 10, lines 6 – 7, and equation 1.

We think that the current figure 1 provides the answer to reviewer's question: Is it the same distribution (mean, rmse). Specifically, Figure 1 shows that the available TEC data values are in most cases close to the interpolated TEC data values. Therefore, in

order to save space, we may not present the figure below which is required by the reviewer in this comment.

The available 121447 TEC data histogram is as shown in the top panel of Figure below. The bottom panel of the figure shows that of finally interpolated 331,776 TEC data.

[Figure]

It can be seen in the above figure that the bottom panel does not have the spikes visible in top panel which appear like outliers. The patterns of distribution of data in the two panels appear to be similar.

**Comment**:

(m) p.13, line 13. The authors justify the quality of their model by the existence of a secondary peak at the magnetic equator already observed elsewhere. However, if I make a vertical line around 16 LT for example on the observed or on the model (Figure 1), I will see an irregular variation of the TEC (~ 10 tecu) in latitude with many secondary peaks (southern hemisphere for example) and not a steady decrease as

expected. These secondary peaks are not physical and are due to averaging (hence my question about variability in a cell) and to interpolation. What do the authors think?

Response:

The statement on page 13, lines 24 – 26 where we mentioned "…validation using data that was not included during modeling is provided in section 5" implies that Figure 2 was presented not solely to justify the quality of our model. Although not explicitly stated, another aim of Figure 2 was to demonstrate that the binned data used during our model development exhibits the known ionospheric TEC features. Indeed, at specific locations within low-latitude regions, the diurnal TEC is known to exhibit a secondary TEC peak produced by a physical process known as Pre Reversal Enhancement (PRE). Therefore, the observation of a secondary TEC enhancement in this study is not associated with measurement or averaging errors. However, we do not dismiss the fact that there could be errors in measurements / averaging as stated by the reviewer.

**Comment**:

(m) p.14, line 11-20. I do not see quite the same thing that the authors describe in particular for the graph c in strong solar activity. Maximum north is on the equator (the 2 bubbles red colored) when expected at 20 ° N? The south EIA maximum appears to be well positioned.

Response:

We acknowledge that the available data might have limited depicting abilities for the exact location of the northern crest of EIA. This issue can be appreciated by comparing the ionospheric features depicted in Figure 2, panels in row (c) with those in Figure below panel (b). The maps of electron density at 100 km altitude (figure below) were presented as Figure 3 in Mungufeni et al (2018): Statistical analysis of the correlation between the equatorial electrojet and the occurrence of the equatorial ionization anomaly over the East African sector, Ann. Geophys., 36, pp. 841 – 853, 2018.

In figure below, panel (b), the trough of EIA appears to be shifted south of the magnetic equator. This observation is consistent with that on Figure 2, panels in row (c) where the

trough appears to be shifted south of the magnetic equator. Therefore, the magenta lines in Figure 2, panels of row (c) might pass over the inner wall of the northern crest.

[Figure]

**Comment**:

(m) p.16, line 10. A first evaluation is made only on longitude 37.5 ° E due to the existence of GPS measurements and publications around these measurements. Fortunately, there are African stations at other longitudes. My question: since the model is built between -20 and 60° E longitude, are the conclusions of 37.5 °E longitude valid for other longitudes? I would have seen an overall statistical result but I have no idea because the difference (observedmodeled) is less than 0.1 tecu on the 2 examples. Is this same conclusion for all longitudes?

Response:

The discussions along 37.5° E longitude referred to by the reviewer involve (i) equinoxial asymmetry of TEC, (ii) occurrence of lowest TEC in June solstice, and (iii) high values of TEC in December. Since figure 8 in the manuscript depicts the 3 discussion points, our answer to the question posed by the reviewer (since the model is

built between -20 and 60° E longitude, are the conclusions of 37.5 °E longitude valid for other longitudes?) is yes. However, we may not generalize the small error values (<0.1 TECU) for other longitudes.

**Comment**:

(m) p.18, line 10. The authors do not provide any positioning on the 1600 points with an absolute difference modeling of at least 10 tecu. It is certainly for the year 2012 and not 2018 but the points relate to a particular hour or month? [I already pointed out a possible divergence of the model in one of my previous remarks in the case of an extrapolation with solar activity].

Response:

On page 20, lines 9 – 11, we have stated that the high errors maybe partly attributed to the limitation of spline modeling technique (inability to extrapolate), discussed in subsection 5.1.

**Comment**:

(M) p.21, lines 14-16. The authors validated their model with ionosondes in South Africa, therefore located in mid-latitudes. I think it is an exaggeration to say that we would have the same result ('predicted **fairly well** using our model.') With a low latitude ionosonde, the study remains to be done!

Response:

As shown on page 23, line 15, we have now specified that the "fairly well" is based on validation using mid-latitude stations. Moreover, we have also stated that we might validate our model over low-latitude region that falls within the current study area when in future ionosonde observations become available over the region. See page 23, line 16 and page 24, lines 1 - 2.

**Comment**:

(M) p.25, line 4. I do not agree with this conclusion. It's because the TEC variations are more irregular with the spline model compared to the NN model that it is the best!

Admittedly, the variations of TEC with NN are over-smoothed (but GIM / CODG also for example) but the many variations in Figure 7 are first linked to the error on the profile estimated by RO and by the procedures of interpolation to give values the nodes of the grid which is a **mathematical** filling and not a **physical** one.

Response:

In the text which is being referred to (see page 26, lines 8 – 11, page 27, lines 1 - 3), we only provided the difference between output of our model and that of neural network. We did not make judgment that ours is the best. However, we attempted to explain why our model output is irregular.

The text being referred to is copied and pasted below.

"Another observation that can be made from Figures 7 and 8 is that unlike the neural network model which yields smooth spatial TEC variation, the spline modeling technique does not yield smooth spatial TEC variation. In real life, measurement or observed values rarely vary smoothly. Since the spline modeling technique produces results (see Figure 1) which demonstrate that the modeled data matches almost perfectly the observed data, it is expected that the spatial variations of TEC in maps of Figure 7 are not smooth."

**Comment**:

I also regret that the comparison of Figures 6 and 7 is purely visual and that there are no statistical figures of differences in the proposed text.

Response:

The text on page 24, lines 5 – 12 justifies the purely visual comparison of Figures 7(old 6) and 8 (old 7). We have copied and pasted the text below.

"It would be good to compare error levels produced when some measured TEC are compared with modeled TEC generated by (i) the existing regional TEC models

discussed in section 1 and (ii) our spline technique TEC model. We may not perform such analysis since models in (i) are based on electron density integrated from ground up to GPS satellites (~20,200 km), while model in (ii) is based on electron density integrated up to ~800 km. However, we present Figures 7 (old 6) and 8 (old 7) to compare EIA features captured by our spline technique model with those by the neural networks technique of Okoh et al., (2019)."

**Comment**:

My conclusion is that there is a **real work of exploiting the RO data** for modeling purposes. The initial difficulty is the lack of measures to cover the Africa zone. Also, the authors were forced to introduce mathematical approaches to cover all the variables retained. They justified their model on a physical result of maps reproducing the large known variability's. The model does not allow a fine-grained approach to the ionosphere compared to a more regional modeling with GNSS measurements. Their current conclusion is that their model leads to better results than the 2 empirical models (IRI and NeQuick) widely used. If the authors want to see their results applied to future studies, they must publish the coefficients of their model. Is this an objective of the authors?

Response:

Once again, we thank the reviewer for recognizing the enormous work done in this manuscript. A decision about publishing the developed model coefficients will be taken later. However, we can avail to anyone on request, particularly for educational purposes.